# Human GBP1 is a microbe-specific gatekeeper of macrophage apoptosis and pyroptosis

Daniel Fisch[1,2] (ID), Hironori Bando[3,4], Barbara Clough[1], Veit Hornung[5], Masahiro Yamamoto[3,4], Avinash R Shenoy[2,6,*] (ID) & Eva-Maria Frickel[1,**] (ID)

## Abstract

The guanylate binding protein (GBP) family of interferon-inducible GTPases promotes antimicrobial immunity and cell death. During bacterial infection, multiple mouse Gbps, human GBP2, and GBP5 support the activation of caspase-1-containing inflammasome complexes or caspase-4 which trigger pyroptosis. Whether GBPs regulate other forms of cell death is not known. The apicomplexan parasite *Toxoplasma gondii* causes macrophage death through unidentified mechanisms. Here we report that *Toxoplasma*-induced death of human macrophages requires GBP1 and its ability to target *Toxoplasma* parasitophorous vacuoles through its GTPase activity and prenylation. Mechanistically, GBP1 promoted *Toxoplasma* detection by AIM2, which induced GSDMD-independent, ASC-, and caspase-8-dependent apoptosis. Identical molecular determinants targeted GBP1 to *Salmonella*-containing vacuoles. GBP1 facilitated caspase-4 recruitment to *Salmonella* leading to its enhanced activation and pyroptosis. Notably, GBP1 could be bypassed by the delivery of *Toxoplasma* DNA or bacterial LPS into the cytosol, pointing to its role in liberating microbial molecules. GBP1 thus acts as a gatekeeper of cell death pathways, which respond specifically to infecting microbes. Our findings expand the immune roles of human GBPs in regulating not only pyroptosis, but also apoptosis.

**Keywords** apoptosis; caspases; pyroptosis; *Salmonella* Typhimurium; *Toxoplasma gondii*
**Subject Categories** Immunology; Microbiology, Virology & Host Pathogen Interaction
**The EMBO Journal (2019) 38: e100926**

## Introduction

Interferon gamma (IFNγ) plays a pivotal role in antimicrobial defense, and loss-of-function mutations in IFNγ-receptor signaling in the human and mouse result in severe susceptibility to infections, including by bacteria and microbial parasites (Altare *et al*, 1998; Levin & Newport, 1999; Janssen *et al*, 2002). IFNγ-priming of macrophages, which is crucial in restricting pathogen growth and controlling infection, upregulates key antimicrobial molecules that mediate cell-intrinsic control of pathogens (Mosser & Edwards, 2008; MacMicking, 2012; Randow *et al*, 2013) and enhances pyroptotic cell death (Shenoy *et al*, 2012; Meunier *et al*, 2014; Pilla *et al*, 2014; Finethy *et al*, 2015; Santos *et al*, 2018).

Among the most abundant IFNγ-induced immune proteins are the guanylate binding proteins (GBPs), a family of large GTPases (65–72 kDa) which restrict bacterial replication and promote inflammasome signaling (Kim *et al*, 2011; Shenoy *et al*, 2012; Meunier & Broz, 2016; Pilla-Moffett *et al*, 2016; Ngo & Man, 2017). The human genome harbors 7 *GBP* genes, among which GBP1-5 are the most ubiquitously expressed members and GBP6 and 7 are only found in epithelial cells lining mucosal surfaces of the lung and intestines (Uhlen *et al*, 2015). In contrast, the mouse genome encodes 11 *Gbps* at two separate chromosomal loci (chr 3 and chr 5), whose protective roles have been revealed through studies on mice lacking *Gbp1, Gbp2, Gbp5*, or all *Gbps* on Chr3 (Degrandi *et al*, 2007, 2013; Kim *et al*, 2011; Shenoy *et al*, 2012; Yamamoto *et al*, 2012; Lindenberg *et al*, 2017). Mouse mGbp1, 2, 3, 5, 6, 7, 9, and 10 can be recruited to vacuoles of bacteria such as *Salmonella, Listeria, Francisella, Mycobacterium, Chlamydia,* and *Legionella* (Kim *et al*, 2011; Degrandi *et al*, 2013; Haldar *et al*, 2014; Meunier *et al*, 2014, 2015; Feeley *et al*, 2017; Finethy *et al*, 2017; Lindenberg *et al*, 2017) or parasites such as *Toxoplasma gondii* (*Tg*; Degrandi *et al*, 2007; Virreira Winter *et al*, 2011; Kravets *et al*, 2012, 2016; Selleck *et al*, 2013; Haldar *et al*, 2015). Human GBP1, 2, and 5 have a C-terminal CaaX motif that can be isoprenylated to facilitate membrane recruitment (Britzen-Laurent *et al*, 2010). Additionally, GTPase activity

---

1  Host-*Toxoplasma* Interaction Laboratory, The Francis Crick Institute, London, UK
2  MRC Centre for Molecular Bacteriology & Infection, Imperial College, London, UK
3  Department of Immunoparasitology, Research Institute for Microbial Diseases, Osaka University, Osaka, Japan
4  Laboratory of Immunoparasitology, WPI Immunology Frontier Research Center, Osaka University, Osaka, Japan
5  Gene Center and Department of Biochemistry & Center for Integrated Protein Science (CIPSM), Ludwig-Maximilians-Universität München, Munich, Germany
6  The Francis Crick Institute, London, UK
   *Corresponding author. Tel: +44 2075943785; E-mail: a.shenoy@imperial.ac.uk
   **Corresponding author. Tel: +44 2037961438; E-mail: eva.frickel@crick.ac.uk

and GBP oligomerization, which depend on critical amino acids, have been studied biochemically before and need to be characterized in infected human macrophages (Daumke & Praefcke, 2016; Shydlovskyi *et al*, 2017). Murine Gbps facilitate the release of microbial ligands, such as LPS or DNA, and promote pyroptosis (Meunier *et al*, 2014, 2015; Pilla *et al*, 2014; Haldar *et al*, 2015; Man *et al*, 2015; Feeley *et al*, 2017; Finethy *et al*, 2017; Piro *et al*, 2017). However, mGbp2 can promote LPS detection by caspase-11 independently of bacterial lysis (Pilla *et al*, 2014). Similarly, GBP5/mGbp5 has been implicated in the regulation of NLRP3 activation by pathogenic bacteria and soluble inflammasome priming agents (Shenoy *et al*, 2012), and caspase-11-driven responses to LPS-containing outer membrane vesicles (OMVs; Santos *et al*, 2018). Strikingly, how human macrophages are equipped to defend themselves against bacterial or parasitic pathogen infections, via the breadth of roles the human GBPs play, remains unstudied. Recent studies have shown cell type-specific roles of human GBPs which underscore the need for defining the roles of GBPs in macrophages during infection (Al-zeer *et al*, 2013; Johnston *et al*, 2016; Qin *et al*, 2017).

Inflammasomes are multimolecular complexes that activate caspase-1 in response to microbial or danger/damage-associated signals (Broz & Dixit, 2016). In specific contexts, such as infection by Gram-negative bacteria, caspase-4 and caspase-8 can regulate the assembly of inflammasomes and caspase-1 activation (Kayagaki *et al*, 2011; Rathinam Vijay *et al*, 2012; Man *et al*, 2013; Gurung *et al*, 2014; Antonopoulos *et al*, 2015; Viganò *et al*, 2015). The release of *Francisella* DNA by mGbp2 and mGbp5 is detected by the DNA-binding protein AIM2, which activates caspase-1 in mouse macrophages through the adaptor protein ASC (Man *et al*, 2015; Meunier *et al*, 2015). However, recent studies in primary human myeloid cells have suggested that cytosolic DNA stimulates a cGAS-STING-dependent pathway that results in secondary NLRP3 activation, whereas AIM2 was dispensable (Fernandes-Alnemri *et al*, 2009; Gaidt *et al*, 2017).

IFNγ-signaling deficiency in mice results in extreme susceptibility to infection by the eukaryotic pathogen *Tg* and the Gram-negative bacterial pathogen *Salmonella enterica* Typhimurium (STm), among others (Jouanguy *et al*, 1999; Hunter & Sibley, 2012). *Tg* infection causes chronic disease that can lead to death in the immunocompromised and fetal abnormalities in cases of the mother acquiring a primary infection during pregnancy (Pappas *et al*, 2009; Torgerson & Mastroiacovo, 2013). STm is a broad host-range pathogen that causes gastrointestinal disease in humans (Coburn *et al*, 2007). Human IFNγ deficiency also results in sensitivity to environmental salmonellae (Jouanguy *et al*, 1999), pointing to a need to better understand the response of human cells and IFNγ-mediated responses to these pathogens.

Both pathogens can induce cell death in the infected cell, which may promote the spread of infection within the host (Coburn *et al*, 2007; Krishnamurthy *et al*, 2017). *Tg*-induced host-cell death can vary depending on the host species and IFNγ-priming; mechanisms including pyroptosis and inhibition of apoptosis have been reported in *Tg*-infected naïve rodent and human cells. However, the response of human IFNγ-primed macrophages to *Tg* remains unclear. STm activates the NLRC4 and NLRP3 inflammasomes in mouse and human macrophages (Broz *et al*, 2010; Man *et al*, 2014; Kortmann *et al*, 2015; Reyes Ruiz *et al*, 2017). Further, IFNγ-priming of human

macrophages enhances STm-induced inflammasome activation (Shenoy *et al*, 2012; Meunier *et al*, 2014; Santos *et al*, 2018). Whether other GBPs contribute to inflammasome responses to STm or *Tg* remains to be determined.

In this study, we systematically studied the roles of human GBPs in primary monocyte-derived macrophages (MDMs) and PMA-differentiated THP-1 cells infected with *Tg* and STm. Notably, *Tg* infection in macrophages caused GBP1-dependent apoptosis and STm infection led to GBP1-dependent increase of pyroptosis. Our studies uncover a gatekeeping role for human GBP1 and expand the role of human GBPs in regulating other forms of cell death during natural infection by microbial pathogens.

# Results

## GBP1 is an essential mediator of macrophage cell death during *Toxoplasma* infection

We investigated the impact of IFNγ-priming on host cell death in PMA-differentiated human THP-1 macrophage-like cells upon infection with type I (RH) and type II (Pru) *Tg*, which differ in their virulence effector repertoire (Saeij *et al*, 2005). We found that 24 h post-infection with both *Tg* strains IFNγ-primed macrophages underwent enhanced cell death, as measured by lactate dehydrogenase (LDH) and XTT dye assays (Fig 1A). We hypothesized that similar to their role in murine cells, GBPs could be involved in IFNγ-enhanced macrophage death. Primary MDMs and THP-1 cells treated with IFNγ express GBP1-5 but not GBP6 or 7 (Fig EV1A), and both macrophage types also express low, but detectable, levels of GBP1 in the absence of IFNγ stimulation (Fig EV1B). We silenced individual GBP1–5 by siRNA transfection (Fig EV1C) and quantified type I and type II *Tg* infection-induced cell death (Fig 1B). Silencing of GBP1, but not other family members, abrogated *Tg*-induced cell death suggesting an essential role for GBP1 in this process. GBP1 silencing in primary MDMs also abrogated *Tg*-induced cell death further validating its role in human macrophages (Fig 1C and Appendix Fig S1A).

*GBP1*-knockout THP-1 (ΔGBP1) cells created with CRISPR/Cas9 (Fig EV1D), also failed to undergo cell death upon *Tg* infection and provided an independent verification of siRNA experiments (Fig 1D). THP-1 ΔGBP1 cells were extensively characterized, and GBP2-5 expression and sequences in THP-1 ΔGBP1 were similar to wild-type THP-1 cells (Fig EV1D–G). Genetic complementation of THP-1 ΔGBP1 cells was achieved using a Doxycycline (Dox)-inducible expression system which mimicked IFNγ-induced expression of GBP1 (Fig EV2A–C). GBP1 protein abundance remained high over a 24-h period with a protein half-life of 6.3 h, which permitted Dox pre-treatment and removal during the course of experiments (Fig EV2D). Indeed, re-expression of GBP1 (THP-1 ΔGBP1 + Tet-GBP1 cells) restored cell death in response to *Tg* infection, which provided additional support for siRNA and CRISPR/Cas9-based experiments and underscored the role of GBP1 in cell death (Fig 1D). Real-time propidium iodide (PI) uptake assays, measuring membrane damage linked to cell death, revealed the kinetics of *Tg*-induced cell death (Fig 1E). These assays confirmed the requirement for IFNγ-priming and the initiation of membrane damage ~ 6 h post-*Tg* infection (Fig 1E). Similarly, Dox-induced expression

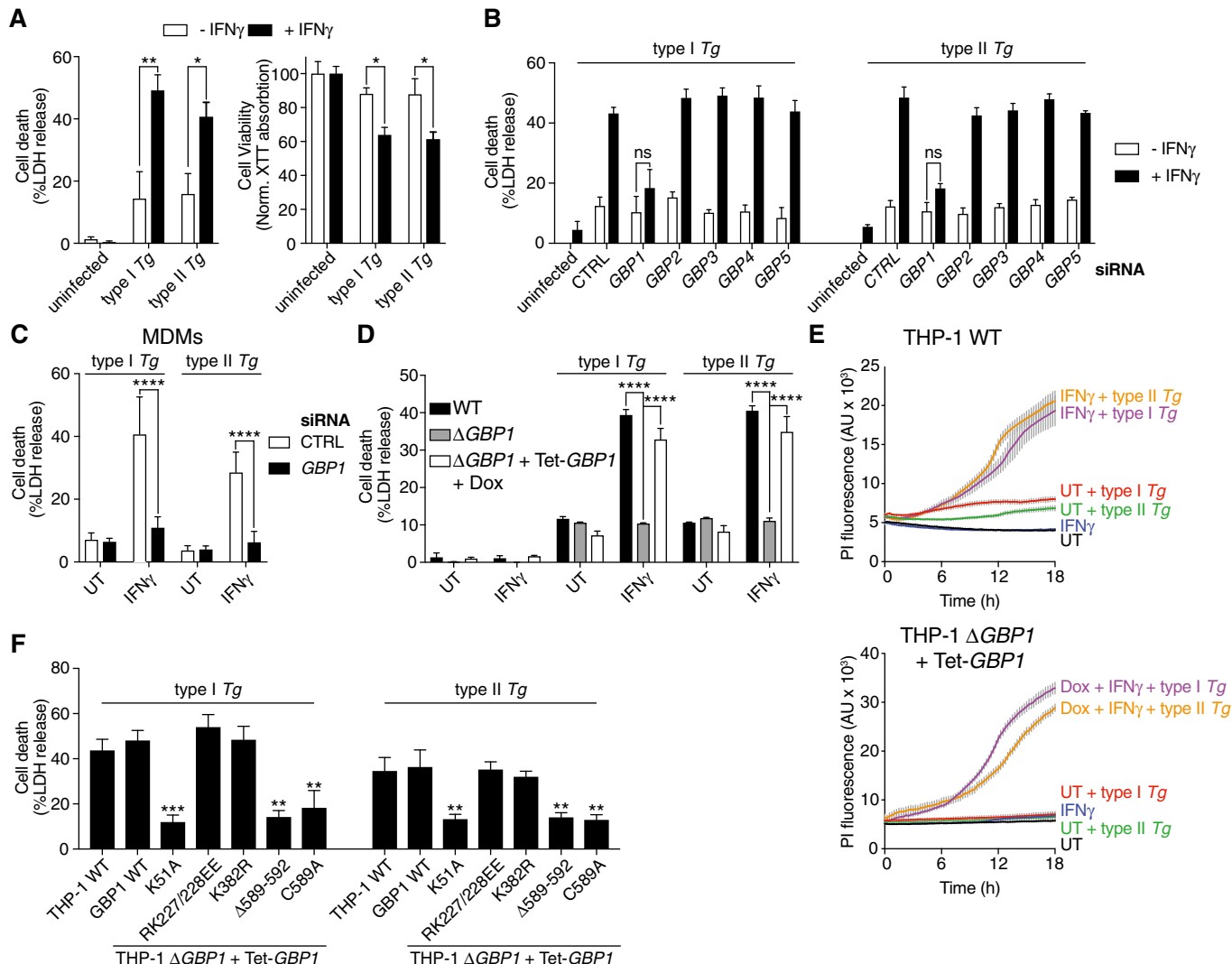

**Figure 1. GBP1, its GTPase activity, and isoprenylation are required for IFNγ-enhanced cell death upon *Toxoplasma* infection.**

A   IFNγ enhances macrophage host cell death after type I (RH) and type II (Pru) *Toxoplasma gondii* (*Tg*) infection. Graphs show cell death as measured by LDH release assay (left) and viability measured by XTT assay (right) in untreated or IFNγ-primed THP-1 cells infected with type I or type II *Tg* for 24 h.

B   LDH release assays from THP-1 cells left untreated or primed with IFNγ, transfected with siRNA against indicated *GBPs* or non-targeting control (CTRL), and infected with indicated strains of *Tg* for 24 h.

C   LDH release assay from primary monocyte-derived macrophages (MDM) left untreated or treated with IFNγ, transfected with siRNA against *GBP1* or non-targeting control (CTRL), and infected with indicated strain of *Tg* for 24 h. Mean ± SEM of n = 4 donors shown.

D   LDH release assay from indicated wild-type, *GBP1*-knockout (Δ*GBP1*), or *GBP1*-reconstituted (Δ*GBP1* + Tet-*GBP1*) cells infected with type I or type II *Tg* for 24 h. Cells were untreated or treated with IFNγ or additionally treated with Doxycycline (Dox) as indicated.

E   Real-time propidium iodide (PI) uptake assay from the indicated THP-1 cells infected with type I or type II *Tg*, and untreated or treated with IFNγ and/or Dox as labeled. AU, arbitrary units.

F   LDH release assay from wild-type THP-1 or Δ*GBP1* cells stably reconstituted with Dox-inducible expression plasmids of the indicated mutants of GBP1. Cells were pre-treated with IFNγ and Dox and infected with either type I or type II *Tg* for 24 h.

Data information: Graphs in (A, B and D–F) show mean ± SEM from n = 3 independent experiments. *P ≤ 0.05; **P ≤ 0.01; ***P ≤ 0.001; ****P ≤ 0.0001 for indicated comparisons in (A–D) from two-way ANOVA or comparison between THP-1 WT and other genotypes by one-way ANOVA in (F) following adjustment for multiple comparisons; ns, not significant.

of GBP1 was necessary for PI uptake in IFNγ-primed THP-1 Δ*GBP1* + Tet-*GBP1* cells (Fig 1E).

We next sought to determine the molecular determinants of GBP1 involved in cell death and tested this via mutational analyses. We employed the following GBP1 mutants: K51A lacking GTPase

activity, RK227/228EE with constitutive homodimerization (Vöpel *et al*, 2010), K382R lacking a ubiquitination site, and C589A and Δ589–592 lacking the isoprenylation residue or the CaaX box, respectively (Vöpel *et al*, 2010; Shydlovskyi *et al*, 2017; Fig EV2E and F). Notably, GBP1 variants with loss of GTPase activity (K51A) or

isoprenylation (C589A or Δ589–592) were unable to restore *Tg*-induced cell death in Δ*GBP1* cells (Fig 1F). Taken together, these experiments established that human GBP1 was required for macrophage cell death in response to infection by type I or type II *Tg* and its GTPase activity and isoprenylation were essential for this function.

### *Toxoplasma* induces GBP1-, AIM2-, ASC-, and caspase-8-dependent apoptosis in macrophages

*Tg*-induced cell death in human macrophages is poorly characterized. To investigate mechanisms of cell death, we used pan-caspase inhibitors during infection and found that they completely inhibited cell death (Fig EV3A). As GBPs are known to be involved in pyroptosis, and *Tg* infection of rodent macrophages causes pyroptosis via NLRP1 and NLRP3 (Cirelli *et al*, 2014; Ewald *et al*, 2014; Gorfu *et al*, 2014), we reasoned that human macrophages infected with *Tg* might also undergo pyroptosis. However, unexpectedly, cell death in IFNγ-primed THP-1 Δ*CASP1* or Δ*CASP4* CRISPR/Cas9-knockout cells, or THP-1 with stable silencing of gasdermin D (GSDMD; *GSDMD*^miR cells), the pore forming protein responsible for pyroptosis, was comparable to that in wild-type THP-1 cells (Fig 2A). Maturation of caspase-1 and IL-1β or IL-18 into their active forms could not be detected in *Tg*-infected macrophages irrespective of the absence or presence of GBP1 (Fig EV3B and C). These findings together ruled out pyroptotic cell death 18 h after *Tg* infection. In contrast, THP-1 Δ*ASC* cells were resistant to *Tg*-induced cell death (Fig 2A). As ASC has previously been implicated in apoptosis, we measured apoptosis using an Annexin V (AnnV)-Glo luminescence assay and performed immunoblots for caspases and the apoptotic substrate PARP. Indeed, IFNγ-primed MDMs infected with *Tg* displayed marked apoptosis within 3 h post-infection, which preceded PI uptake induced by post-apoptotic necrotic membrane damage (Silva, 2010; Fig 2B). Further, primary MDMs transfected with GBP1 siRNA failed to show increased AnnV-Glo upon *Tg* infection (Fig 2B and Appendix Fig S1A). Similarly, wild-type, but not THP-1 Δ*GBP1* cells, showed increased AnnV-Glo in response to *Tg* infection (Fig EV3D). Flow cytometry-based quantification of apoptosis using AnnV staining provided concordant results showing elevated apoptosis upon IFNγ-priming and lack of apoptosis in Δ*GBP1* cells (Fig EV3E). We used TNFα plus cycloheximide (CHX) or staurosporine as positive controls to induce apoptosis, which not only verified our assays, but also indicated that wild-type and THP-1 Δ*GBP1* cells do not have broad defects in extrinsic or intrinsic apoptosis (Fig EV3F). These findings indicated that GBP1 was specifically involved in microbe-dependent apoptosis. Immunoblots from MDMs and THP-1 cells infected with type I and type II *Tg* revealed the presence of cleaved caspase-3, caspase-6, caspase-7 and caspase-8, and cleaved PARP (Figs 2C and EV3G). Live time-lapse imaging showed a temporal increase of caspase-3/7 activity, shrinking of the cells, and the presence of apoptotic bodies upon infection with type I or type II *Tg* (Fig 2D and E). The presence of nuclear fragmentation and condensation of chromatin further supported apoptotic cell death during *Tg* infection (Fig EV3H). These experiments unequivocally established that *Tg*-infected human macrophages undergo apoptosis in a GBP1- and ASC-dependent manner.

We next asked which caspase mediated apoptosis in *Tg*-infected macrophages? We therefore performed a family-wide RNAi screen

against caspases to assess their involvement. Silencing of caspase-8 or caspase-10 in THP-1 cells reduced *Tg*-induced apoptosis (Fig 2F and Appendix Fig S1B). Although caspase-8 was also required for *Tg*-induced apoptosis in primary MDMs, caspase-10, which is a human-specific caspase, was dispensable (Fig 2G and Appendix Fig S1A). Silencing of caspase-8 also inhibited cell death in *Tg*-infected THP-1 Δ*CASP1* and Δ*CASP4,* which ruled out a role for these caspases in the process (Fig EV4A). We therefore concluded that caspase-8 was essential for apoptosis upon *Tg* infection. To investigate the potential role of an NLR/ALR that could detect *Tg* infection and induce apoptosis via the adaptor ASC, we used RNAi against common NLRs and AIM2. RNAi experiments in THP-1 cells revealed that silencing of AIM2, but not NLRP1, NLRP3, or NLRC4, abolished apoptosis (Fig 3A and Appendix Fig S1B). AIM2 silencing also led to diminished apoptosis in *Tg*-infected primary MDMs (Fig 3B and Appendix Fig S1A). We therefore concluded that *Tg*-infected human macrophages undergo *GBP1-, AIM2-, ASC-,* and *CASP8*-driven apoptosis.

AIM2 forms oligomeric inflammasome "specks" for caspase-1 activation. Indeed, ~ 20% of *Tg*-infected macrophages had ASC specks; however, 95% of these also recruited caspase-8 (Fig 3C). Like Tg-induced apoptosis, the formation of ASC–caspase-8 specks was dependent on IFNγ treatment and independent of caspase-1 (Fig 3C). Next, we wanted to investigate why AIM2 and ASC did not activate caspase-1 as is typical of AIM2 activation. Immunoblots of *Tg*-infected MDMs (Fig 3D) and THP-1 (Fig EV3I) cells revealed loss of caspase-1, NLRP1, NLRP3, NLRC4 proteins over a time course of 6 h. In agreement with reduced NLRP3 abundance, *Tg*-infected cells responded poorly to the NLRP3 inflammasome-activating toxin nigericin (Fig EV4B). Importantly, the abundance of AIM2, whose expression was upregulated by IFNγ, was unchanged during the course of *Tg* infection. These results also implied that without IFNγ-priming, because of the absence of AIM2 expression, GBP1 expression alone would not be sufficient for cell death. We tested this experimentally by treating THP-1 Δ*GBP1* + Tet-GBP1 cells with Dox to induce GBP1 expression and infected them with *Tg*. Indeed, heterologous GBP1 expression without IFNγ-priming did not restore apoptosis in THP-1 Δ*GBP1* + Tet-GBP1 cells (Fig EV4C). In conclusion, these results showed that GBP1 mediated atypical apoptosis via AIM2-ASC-caspase-8 in *Tg*-infected human macrophages.

### GBP1 is recruited to pathogen-containing vacuoles in a GTPase activity and isoprenylation-dependent manner

Whether human GBP1 targets microbial vacuoles in macrophages is not known. Murine Gbps in general and human GBPs in some cell lines are recruited to bacterial and *Tg* vacuoles (Coers, 2017; Man *et al*, 2017; Saeij & Frickel, 2017; Santos & Broz, 2018). As we identified atypical apoptosis via GBP1 and AIM2, we asked whether GBP1 was recruited to *Tg* parasitophorous vacuoles (PVs) and whether this was required for apoptosis. Endogenous GBP1 in MDMs was recruited to *Tg* PVs in an IFNγ-dependent manner (Fig 4A and B). Similarly, Dox-inducible expression of mCherry (mCH)-tagged GBP1 in THP-1 Δ*GBP1* targeted to the PV upon IFNγ treatment (Fig 4A and B). We also tested whether the mutants GBP1^K51A, GBP1^C589A, and GBP1^Δ589–592, which did not support apoptosis, targeted to PVs. All three mutants failed to decorate *Tg*

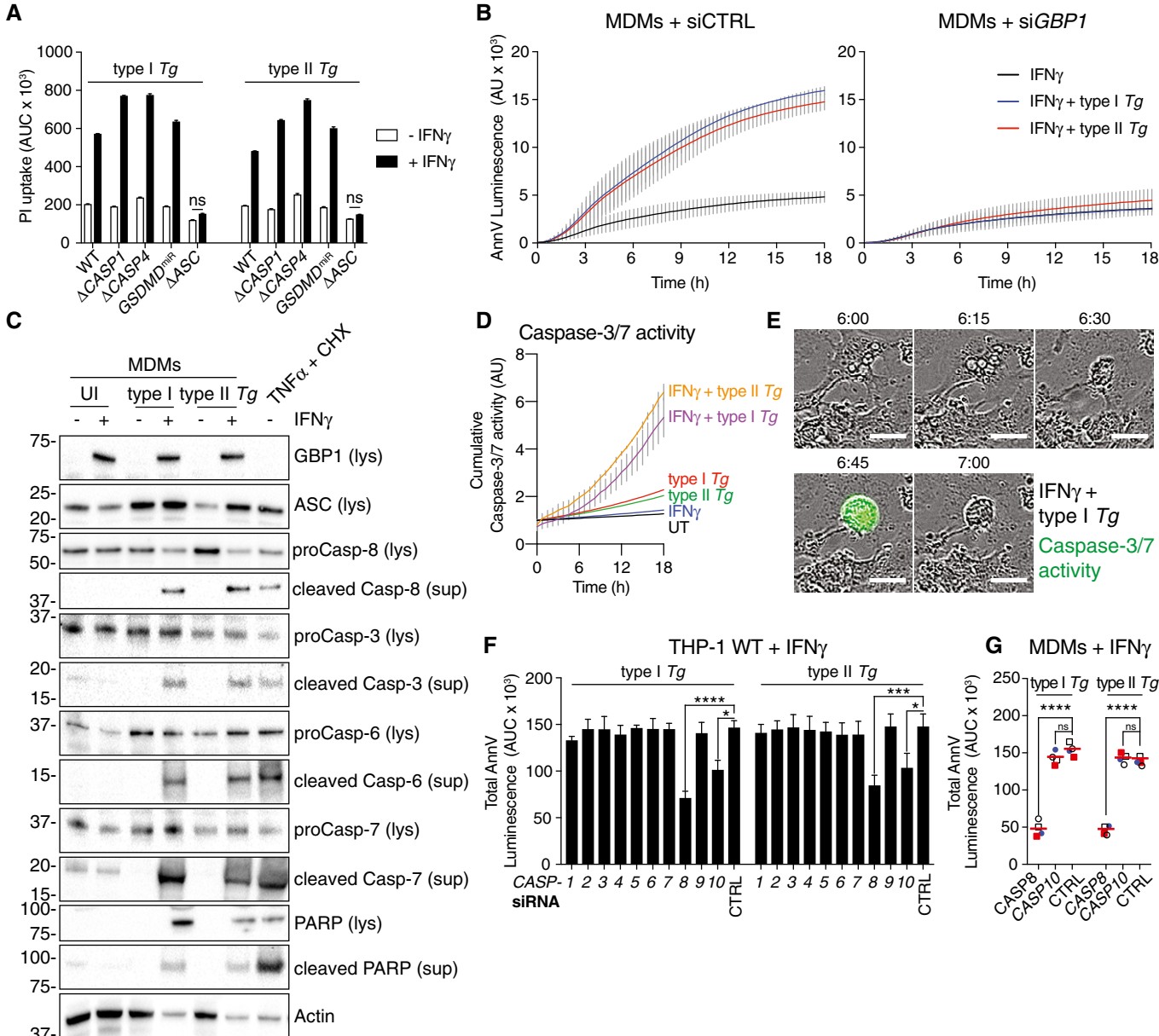

**Figure 2. GBP1 promotes ASC- and CASP8-dependent atypical apoptosis in *Toxoplasma*-infected human macrophages.**

A   Propidium iodide (PI) uptake in wild-type THP-1 or ΔCASP1, ΔCASP4, ΔASC cells, or THP-1 stably expressing *GSDMD*-targeting miRNA (*GSDMD^miR*) infected with the indicated strains of *Toxoplasma gondii* (*Tg*) for 24 h. Cells were untreated or treated with IFNγ as indicated. Mean area under the curve (AUC) ± SEM from *n* = 3 experiments are shown. ns, not significant, from two-way ANOVA following adjustment for multiple comparisons.

B   Real-time apoptosis assay (Annexin V (AnnV)-Glo luminescence) from IFNγ-primed primary human MDMs transfected with non-targeting control (siCTRL; left) or *GBP1* siRNA (si*GBP1*; right) and left uninfected or infected with type I or type II *Tg*. Depicted is the mean ± SEM of *n* = 4 donors. AU, arbitrary units.

C   Immunoblots from primary MDMs infected with the indicated *Tg* strains for 12 h. Cells were untreated or pre-treated with IFNγ before infection. Treatment with TNFα and cycloheximide (CHX) served as positive control for apoptosis. Images are representative of *n* = 4 biologically independent experiments.

D, E Live time-lapse imaging to visualize apoptotic cell death and measure caspase-3/7 activities using a fluorescent caspase substrate. THP-1 WT were primed with IFNγ or left untreated and infected with either type I or type II *Tg*. Plot in (D) shows cumulative caspase-3/7 activity from an 18-h time course (mean ± SEM from *n* = 3 independent experiments). Representative phase-contrast images overlaid with fluorescence signal for caspase-3/7 activity (green) images are shown in (E) from IFNγ-primed THP-1 cells infected with type I *Tg* captured at indicated times post-infection. Scale bar 20 μm.

F   AnnV-Glo assay from IFNγ-primed THP-1 transfected with siRNA against the indicated human caspases and infected with type I or type II *Tg*. Area under the curve (AUC) from a real-time AnnV-Glo luminescence assays similar to those in (B) is plotted as mean ± SEM from *n* = 3 independent experiments. *P ≤ 0.05, ***P ≤ 0.001, ****P ≤ 0.0001 from two-way ANOVA for indicated comparisons following adjustment for multiple comparisons; ns, not significant.

G   AnnV-Glo assay from IFNγ-primed primary human MDMs transfected with siRNA against *CASP8*, *CASP10*, and non-targeting control (CTRL) and infected with type I or type II *Tg*. Area under the curve (AUC) from real-time assays similar to those in (B) are plotted from *n* = 4 independent experiments. Matched shapes and color of symbols indicate donors. ****P ≤ 0.0001 from two-way ANOVA following adjustment for multiple comparisons; ns, not significant.

Source data are available online for this figure.

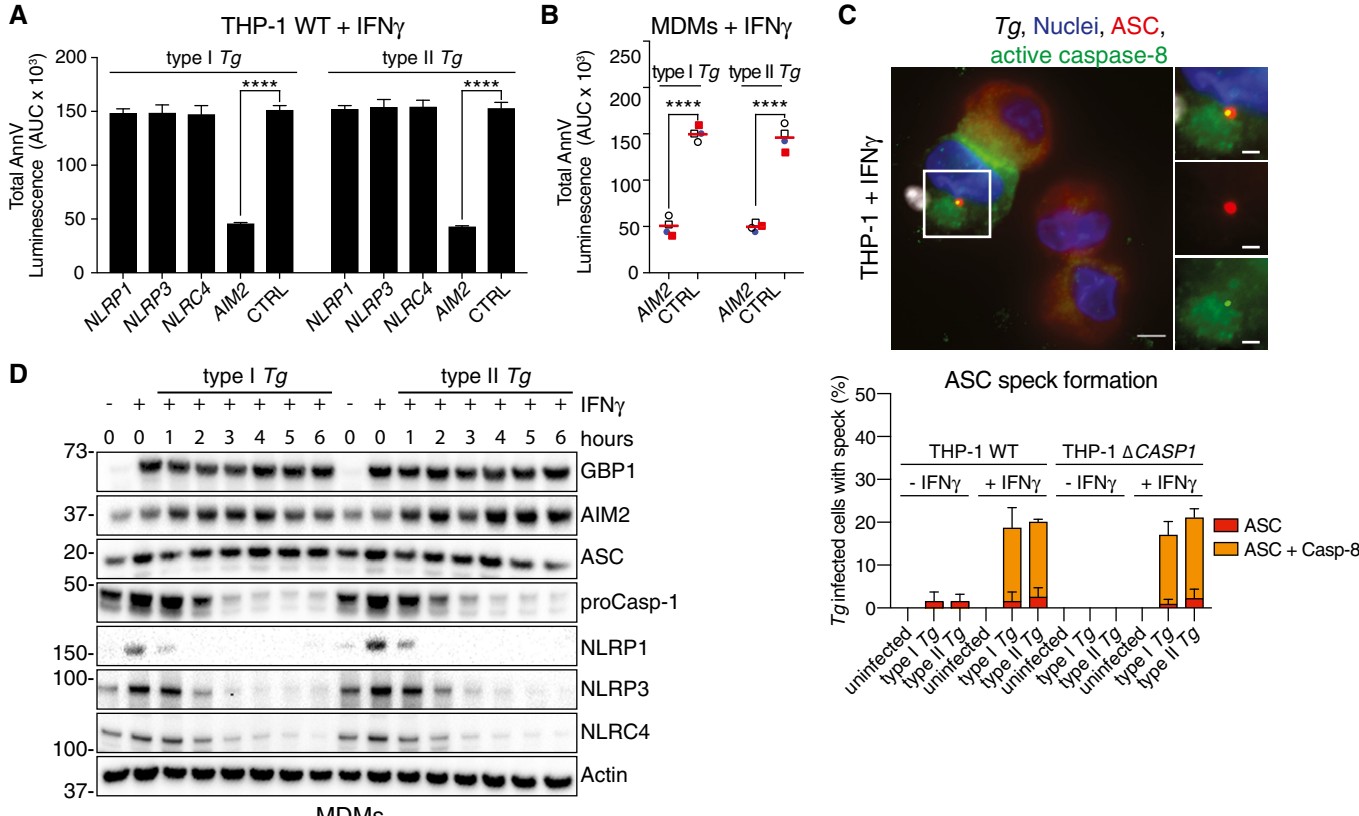

**Figure 3. _Toxoplasma_-infected human macrophages die through AIM2-dependent atypical apoptosis.**

A   AnnV-Glo assay from IFNγ-primed THP-1 transfected with siRNA against the indicated human NLR/ALR sensors and infected with type I or type II _Toxoplasma gondii_ (_Tg_). Area under the curve (AUC) from real-time assays is plotted as mean ± SEM from n = 3 independent experiments. ****P ≤ 0.0001 from two-way ANOVA for indicated comparisons following adjustment for multiple comparisons; ns, not significant.

B   AnnV-Glo assay from IFNγ-primed primary human MDMs transfected with siRNA against _AIM2_ or non-targeting control (CTRL) and infected with type I or type II _Tg_. Area under the curve (AUC) from real-time assays is plotted from n = 4 independent experiments. Matched shapes and color of symbols indicate donors. ****P ≤ 0.0001 from two-way ANOVA for indicated comparisons following adjustment for multiple comparisons; ns, not significant.

C   Representative images (top) from IFNγ-primed THP-1 WT infected with type I _Tg_ for 8 h and stained for ASC and active caspase-8. Blue, Nuclei; Red, ASC; Green, active caspase-8 (IETD-FITC); and Gray, _Tg_. Scale bar, 5 μm or 2 μm in the magnified images. Quantification of ASC specks (bottom) in _Tg_-infected THP-1 WT or ΔCASP1 treated with IFNγ or left untreated at 8 h post-infection, plotted as mean ± SEM from n = 3 independent experiment.

D   Immunoblots from primary MDMs infected with _Tg_ strains for the indicated times. Cells were untreated or pre-treated with IFNγ before infection. Images are representative of n = 4 biologically independent experiments.

Source data are available online for this figure.

PVs, thus correlating with their inability to promote apoptosis (Fig 4A and B). Notably, we did not observe cell death in IFNγ-treated A549 cells infected with _Tg_ (Fig EV4D). We previously showed that in A549 cells, GBP1 does not traffic to _Tg_ vacuoles but is required for restricting _Tg_ replication (Johnston _et al_, 2016). This indicated that even though cell death requires GBP1-targeting of PVs, growth restriction is independent of this process. We therefore propose that PV targeting is required for GBP1-dependent apoptosis during _Tg_ infection.

## GBP1 targets _Salmonella_-containing vacuoles and mediates caspase-4-dependent pyroptosis

To better understand common determinants of GBP1 in influencing host cell death responses to infection, we investigated its role during STm infection. STm infection causes inflammasome-driven pyroptosis in macrophages (Franchi _et al_, 2006; Miao _et al_, 2006; Broz _et al_, 2010). In agreement with previous studies (Shenoy _et al_, 2012), IFNγ-priming of THP-1 cells enhanced pyroptosis and IL-1β release by ~ 1.5–2 fold after infection by STm at various bacterial MOIs (Figs 5A and EV4E–G). Expression of GFP by STm did not affect pyroptosis (Fig EV4E) ruling out non-specific effects of GFP and allowed us to use this strain in our studies which also included immunofluorescence microscopy. Like THP-1 cells, IFNγ-priming of primary MDMs also enhanced STm-induced IL-1β release and pyroptosis (Fig EV4F and G). We tested the involvement of GBP1 in these processes by RNAi in primary MDMs (Appendix Fig S1A) and infection of THP-1 ΔGBP1 cells. In all cases, loss of GBP1 expression abolished IFNγ-mediated enhancement of pyroptosis and reduced it to levels seen without IFNγ-priming (Fig 5B). Together with our results on _Tg_ infection, these findings pointed toward a broader role for GBP1 in host cell death during microbial infections.

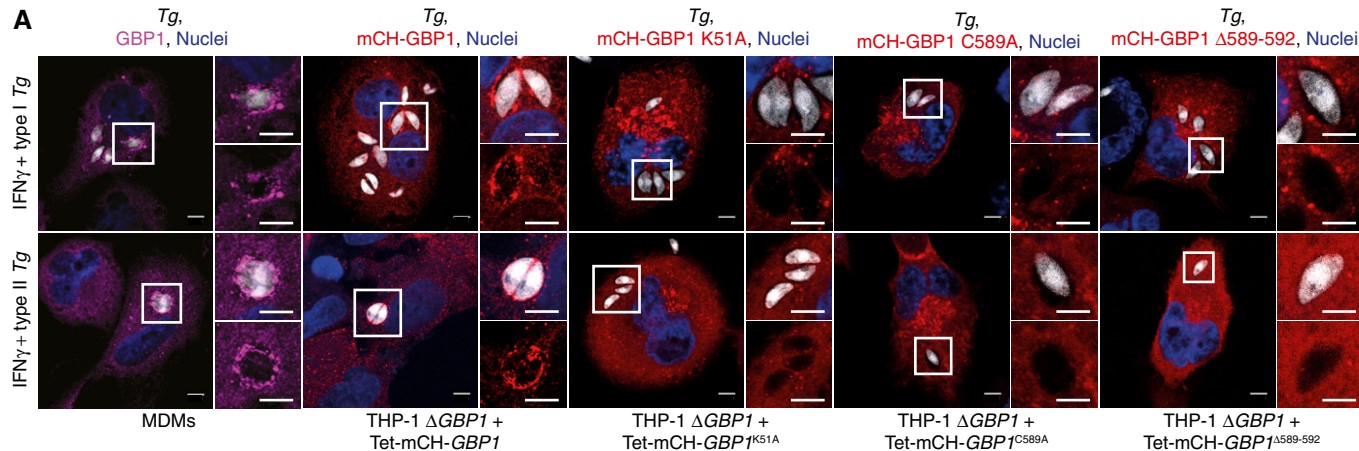

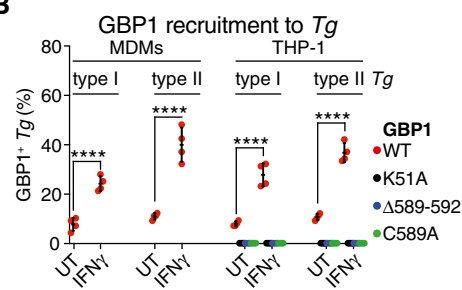

**Figure 4. GBP1 targets *Toxoplasma* vacuoles in human macrophages.**

A, B  Representative images (A) and quantification (B) from IFNγ-primed THP-1 *ΔGBP1* stably reconstituted with Tet-mCH-*GBP1*, mCH-*GBP1*^K51A, mCH-*GBP1*^C589A, or mCH-*GBP1*^Δ589–592 variants or monocyte-derived macrophages (MDMs) stained for endogenous GBP1. Cells were infected with type I or type II *Toxoplasma gondii* (*Tg*) for 2 h. Blue, Nuclei; Gray, *Tg*; Red, mCH-GBP1; or Magenta, GBP1. Scale bar, 5 μm. Graph in (B) shows mean percentage of *Tg* vacuoles targeted by GBP1 from $n = 4$ experiments. ****$P \leq 0.0001$ for indicated comparisons by two-way ANOVA.

Mechanistically, unlike *Tg*, STm-induced cell death was abolished by GSDMD silencing in primary MDMs and THP-1 cells (Figs 5C and EV4H, and Appendix Fig S1A and B), which established that STm-infected naïve and IFNγ-primed macrophages undergo pyroptotic cell death. Importantly, in naïve cells, silencing of caspase-1 or GSDMD reduced STm-induced cell death by ~ 60%, whereas silencing of caspase-4 had no effect (Fig 5C), which indicated that unprimed macrophages infected with STm underwent solely caspase-1-dependent pyroptosis. In contrast, in IFNγ-primed THP-1 and primary MDMs, only the combined silencing of caspase-1 and caspase-4 could completely block STm-induced cell death, which pointed to the involvement of both caspases under these conditions (Figs 5C and EV4I, and Appendix Fig S1A and B). During these experiments, we observed that in IFNγ-primed cells, silencing of either caspase-1 or caspase-4 reduced pyroptosis to levels seen in naïve cells (Fig 5C). We therefore reasoned that IFNγ-induced GBP1 might promote the detection of STm by the noncanonical caspase-4–caspase-1 pathway and enhanced GSDMD processing by both caspases. The existence of a GBP1/caspase-4-dependent pathway was independently supported by experiments in IFNγ-primed THP-1 *ΔCASP1* cells which underwent enhanced STm-induced cell death in response to IFNγ-priming which was abolished upon GBP1 silencing (Fig 5D and Appendix Fig S1B). These findings together indicated that in IFNγ-primed macrophages, STm infection engaged both

caspase-1 and caspase-4 and this required GBP1 (schematic in Fig 5D).

We next asked whether GBP1 GTPase activity and isoprenylation contributed to pyroptosis. Indeed, THP-1 *ΔGBP1* cells expressing GBP1^K51A, GBP1^C589A, or GBP1^Δ589–592 did not undergo STm-induced pyroptosis, whereas no such defects were observed in cells expressing GPB1^RK227/228EE or GBP1^K382R (Fig 5E). Furthermore, endogenous GBP1 translocated to STm-containing vacuoles (SCV) in MDMs (Fig 5 and G). GBP1 expressed in basal levels in MDMs (Fig EV1B) can traffic to SCVs. However, IFNγ-priming, which increased the abundance of GBP1 protein (Fig EV1B), increased the proportion of decorated SCVs by 2-fold (Fig 5G) and also the amount of GBP1 on individual SCVs (Fig 5H). mCH-GBP1, but not GTPase-dead or isoprenylation-deficient variants, recruited to SCVs in THP-1 *ΔGBP1* cells (Fig 5F and G). This suggested that targeting of microbial vacuoles and the same molecular functions of GBP1 were required to mediate apoptosis during *Tg* infection and pyroptosis during STm infection. Importantly, GBP1 did not affect nigericin-induced NLRP3-dependent pyroptosis (Fig EV4J). This indicated that, like in the context of apoptosis, the role of GBP1 was restricted to regulating infection-induced cell death.

We asked whether GBP1 promoted caspase-4 trafficking to SCVs and addressed this using THP-1 *ΔGBP1* + Tet-mCH-*GBP1* cells stably

transduced with catalytically inactive YFP-caspase-4[C258S] (Fig EV4K); stable expression of YFP-caspase-4 led to toxicity. YFP-caspase-4[C258S] translocation could be detected on 37% of SCVs within 2 h of infection in Dox- and IFNγ-primed cells (Fig 5I and J). Remarkably, caspase-4 recruitment was completely absent in cells lacking GBP1 (Fig 5J). Moreover, YFP-caspase-4[C258S] was only recruited to STm that was also decorated with GBP1 (Fig 5J), which resulted in a high Pearson's correlation coefficient for colocalization of GBP1 and caspase-4 on GBP1/caspase-4 double-positive STm (0.76 ± 0.08, $n$ = 529 SCVs). Taken together, we concluded that in

IFNγ-primed cells, GBP1 not only traffics to STm, but also promotes the recruitment and activation of caspase-4.

### GBP1 can be bypassed by direct delivery of *Toxoplasma* lysate or bacterial LPS into the cytosol

Our results showed that GBP1 recruitment to PVs and SCVs correlated with its ability to promote pathogen-dependent cell death. Based on work on murine Gbps (Meunier *et al*, 2014, 2015; Man *et al*, 2015), we hypothesized that GBP1 contributed to the release

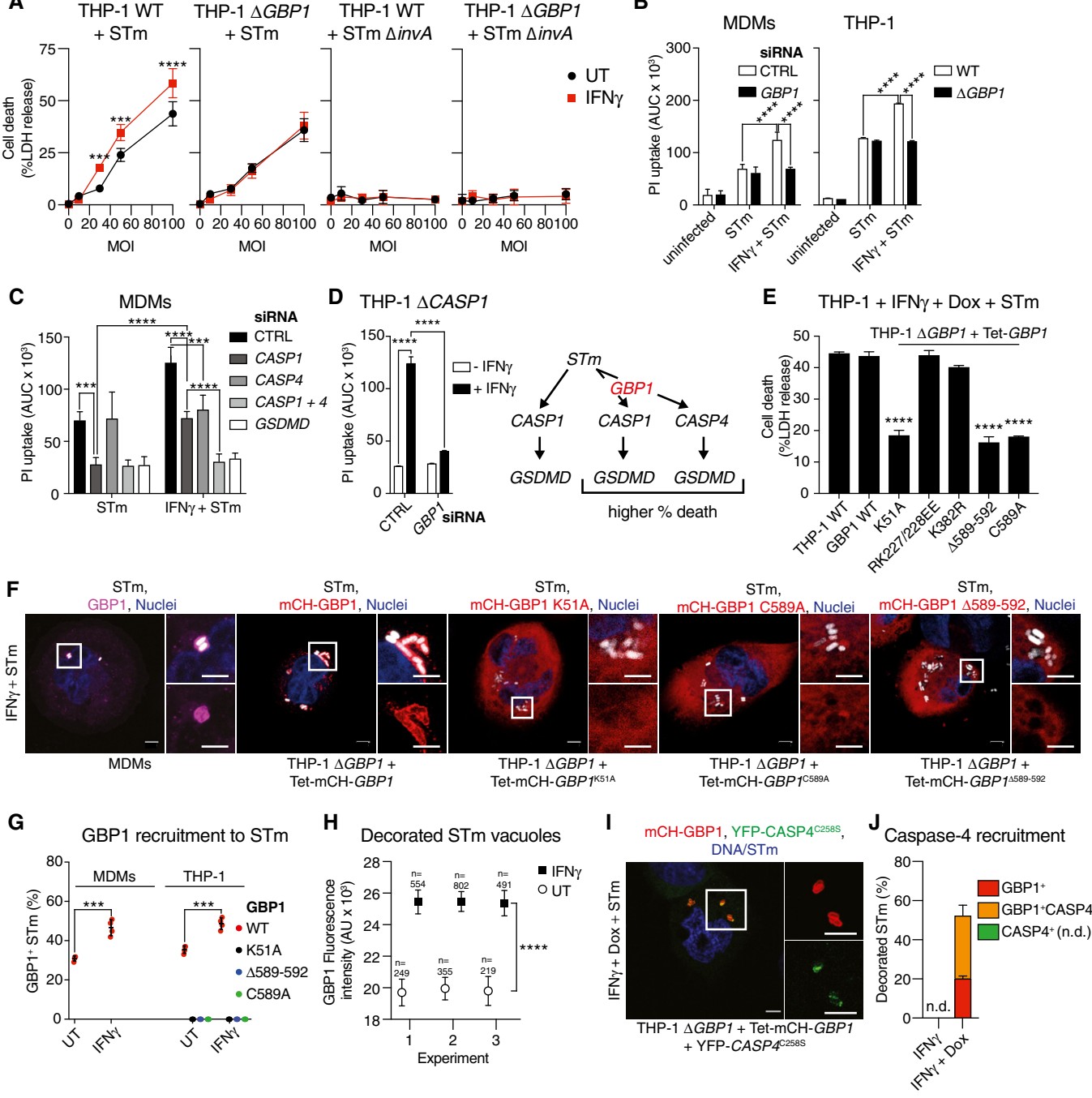

**Figure 5.**

**Figure 5. GBP1 targets *Salmonella* vacuoles and mediates caspase-4-dependent pyroptosis.**

A   LDH release assay from the indicated THP-1 cells untreated or treated with IFNγ and infected with *Salmonella* Typhimurium (STm) WT or a SPI-1 mutant Δ*invA* for 4 h at indicated multiplicities of infection (MOI). Mean ± SEM of *n* = 3 experiments are shown.

B   Propidium iodide (PI) uptake assay from IFNγ-primed primary MDMs transfected with non-targeting control siRNA (CTRL) or siRNA against *GBP1* (left) or indicated THP-1 cells (right) infected with STm-GFP for 4 h. Area under the curve (AUC) from real-time assay plotted as mean ± SEM from *n* = 3 independent experiments.

C   PI uptake assay from IFNγ-primed primary MDMs transfected with indicated siRNA or non-targeting control (CTRL) and infected with STm-GFP for 4 h. Area under the curve (AUC) from real-time assay plotted as mean ± SEM from *n* = 4 independent experiments.

D   PI uptake assay from IFNγ-primed or untreated THP-1 Δ*CASP1* cells transfected with the indicated siRNA and infected with STm-GFP for 4 h. Area under the curve (AUC) from real-time assay plotted as mean ± SEM from *n* = 3 independent experiments. Schematic on right shows overview of the pathway leading to caspase-1 and caspase-4 activation.

E   LDH release assay from the indicated THP-1 cells treated with IFNγ and Dox and infected with STm-GFP for 4 h. Shown is the mean ± SEM of *n* = 3 experiments.

F, G   Representative images (F) and quantification (G) from IFNγ-primed THP-1 Δ*GBP1* cells stably reconstituted with Tet-mCH-*GBP1*, mCH-*GBP1*^K51A, mCH-*GBP1*^C589A, or mCH-*GBP1*^Δ589–592 variants or primary MDMs stained for endogenous GBP1. Cells were infected with STm-GFP for 2 h. Blue, Nuclei; Gray, STm; Red, mCH-GBP1; or Magenta, GBP1. Scale bar, 5 μm. Graph in (G) shows percentage of STm vacuoles targeted by GBP1.

H   Quantification of GBP1 recruitment to STm-GFP-containing vacuoles (SCV) in MDMs pre-treated with IFNγ or left untreated (UT). Mean ± 95% CI of GBP1 fluorescence intensity measured on the indicated number (*n*) of SCVs for each condition from three independent experiments is shown. ****$P < 0.0001$ by nested *t*-test.

I   Representative images from THP-1 Δ*GBP1* + Tet-mCH-*GBP1* cells stably expressing YFP-CASP4^C258S. Cells were infected with STm for 2 h. Blue, Nuclei & STm (Hoechst dye); Red, mCH-GBP1; Green, YFP-CASP4^C258S. Scale bar, 5 μm.

J   Quantification of THP-1 Δ*GBP1* + Tet-mCH-*GBP1* cells stably expressing YFP-CASP4^C258S and infected with STm for 2 h. Graph shows percentage of STm vacuoles targeted by GBP1, CASP4, or both from *n* = 3 experiments. n.d. not detected.

Data information: ***$P ≤ 0.001$, ****$P ≤ 0.0001$ for indicated comparisons by two-way ANOVA in (B–D, G), comparison between untreated and IFNγ-treated in (A) or THP-1 WT, and other genotypes in (E) by one-way ANOVA following adjustment for multiple comparisons; ns, not significant.

of microbial ligands for detection by AIM2 or caspase-4 in the cytosol of human macrophages (Liu *et al*, 2018). To obtain experimental support for this, we asked whether GBP1 could be bypassed by transfecting cells with soluble *Tg* antigen (STAg) or LPS. Indeed, STAg transfection led to IFNγ-dependent cell death as measured by AnnV-Glo in MDMs and THP-1 cells and pointed toward the requirement of an IFNγ-induced factor for sensing STAg ligands (Fig EV5A and B). As levels of caspase-1 were reduced upon *Tg* infection, we transfected STAg in Δ*CASP1* cells to mimic *Tg* infection. Indeed, like natural *Tg* infection, STAg transfection triggered IFNγ-dependent apoptosis in Δ*CASP1* cells (Fig 6A). To identify which component of STAg was required for apoptosis, we treated STAg with DNase, RNase, trypsin, or a mix of deglycosylation enzymes prior to transfection. DNase-treated STAg failed to induce apoptosis in Δ*CASP1* cells and established an essential role of *Tg* DNA in this process (Fig 6B). We further verified this by immunoblots and AnnV-Glo assays to detect apoptosis following STAg transfection of Δ*CASP1* cells silenced for the expression of *AIM2*, *ASC*, *CASP8*, or *GBP1*. These experiments revealed that STAg induced GBP1-independent and AIM2-, ASC-, CASP8-dependent activation of the apoptosis cascade (Fig 6C and D). Taken together, these experiments reconstituted *Tg*-induced apoptosis in Δ*CASP1* cells and pointed to the detection of *Tg* DNA in the cytosol by AIM2. Furthermore, loss of caspase-1 protein during *Tg* infection contributed to the switch in AIM2-dependent pyroptosis to apoptosis.

We performed similar experiments by transfecting LPS to assess the role of GBP1 in promoting caspase-4 activation. LPS transfection of MDMs (Fig 6E) and THP-1 (Fig EV5C) led to GBP1-independent pyroptosis which was in contrast to the role of GBP1 during natural infection with STm. As a control, we verified that IL-1β production by LPS transfection was reduced by GBP5 silencing as has been described before (Shenoy *et al*, 2012; Santos *et al*, 2018; Fig EV5D). Altogether, our results showed that GBP1 acted as a gatekeeper and promoted the release of or access to microbial ligands for cytosolic

receptors such as AIM2 and caspase-4 which trigger pathogen-specific cell death pathways.

# Discussion

Here, we report that human GBP1 acts as an upstream regulator of diverse forms of cell death in macrophages and promotes the activation of microbe-specific downstream pathways. The access to or release of microbial ligands in infected macrophages required GBP1, which therefore acts a gatekeeper of microbe-induced cell death. GBP1 joins GBP2 and GBP5 in regulating human macrophage pyroptosis, but uniquely also regulates the induction of apoptosis. Our results highlight the contribution of IFNγ priming, the host species, and microbial pathogen to macrophage cell death. For instance, in naïve cells, *Tg* induces necrosis in mouse embryonic fibroblasts (Zhao *et al*, 2009), pyroptosis in rat and murine macrophages (Cavailles *et al*, 2014; Cirelli *et al*, 2014; Ewald *et al*, 2014; Gorfu *et al*, 2014), and pyroptosis in human monocytes (Witola *et al*, 2011; Gov *et al*, 2017). In contrast, *Tg* infection of IFNγ-stimulated human foreskin fibroblasts causes an unspecified form of cell death (Gov *et al*, 2013; Niedelman *et al*, 2013). We also found that during STm infection, IFNγ enhanced macrophage pyroptosis in a GBP1-dependent manner. Importantly, both pathogens relied on similar molecular determinants of GBP1 to promote host cell death.

GBP1, GBP2, and GBP5 have C-terminal CaaX motifs that undergo isoprenylation. GBP1 also undergoes GTP-dependent oligomerization (Daumke & Praefcke, 2016). We showed that GBP1 GTPase activity and lipidation are essential for its role in targeting of microbial vacuoles and mediating host-cell death. These findings ascribe additional biological relevance to GBP1 lipidation as demonstrated from *in vitro* experiments with unilamellar vesicles and association with phagosomes (Shydlovskyi *et al*, 2017) and STm in HeLa cells (Bradfield, 2016). A similar role for lipidation of GBP5 for its recruitment to LPS OMVs and mouse Gbp1/2 for trafficking to

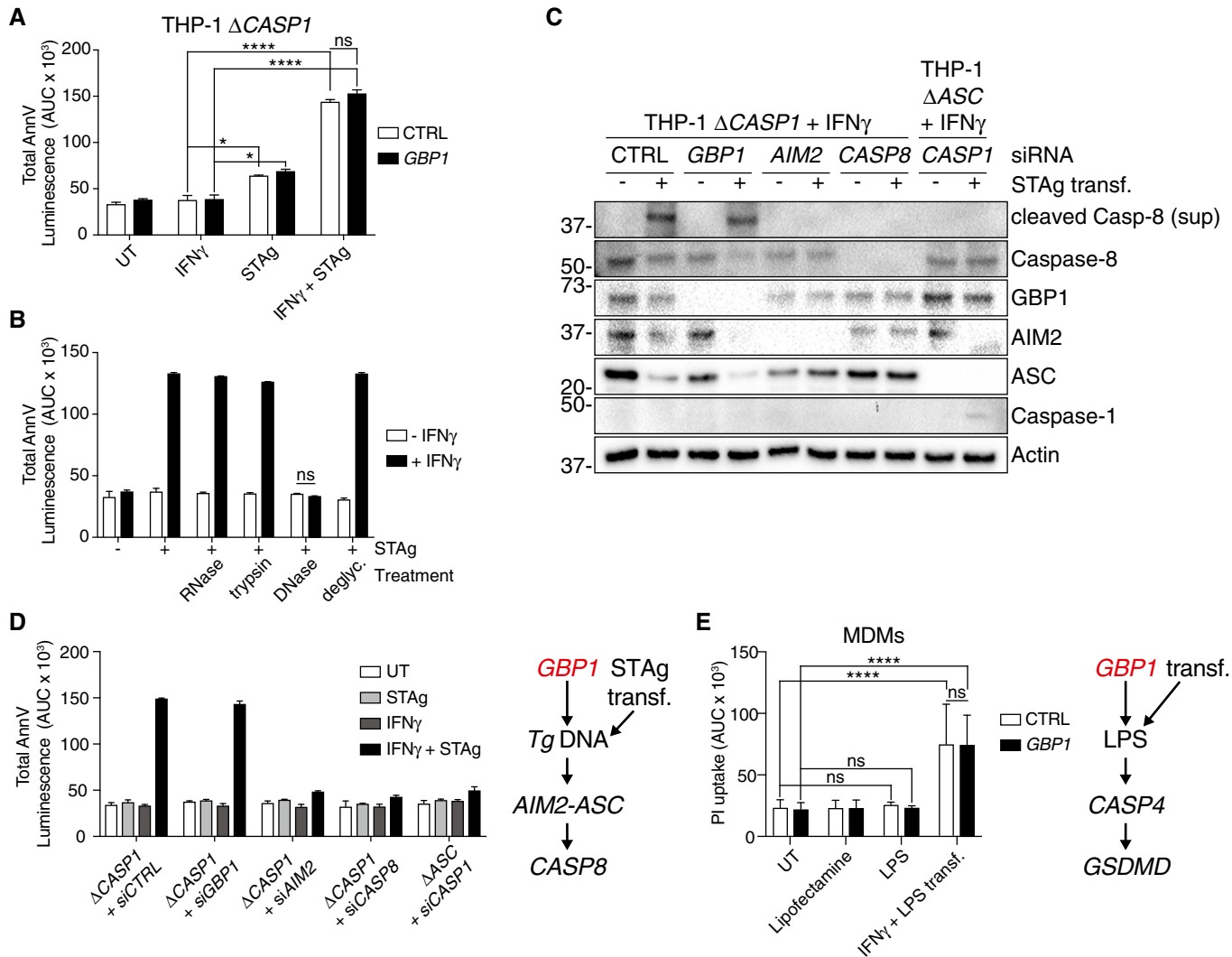

**Figure 6. Transfection of pathogen-derived ligands bypasses GBP1 in inducing macrophage death.**

A  Cell death measured using Annexin V (AnnV)-Glo assay of THP-1 ΔCASP1 cells transfected with non-targeting control (CTRL) or GBP1 siRNA and treated as indicated. Cells were primed with IFNγ, transfected with soluble Tg antigen (STAg) or both, or left untreated (UT). Area under the curve (AUC) from real-time assay plotted as mean ± SEM from n = 3 independent experiments.

B  AnnV-Glo assay from naïve or IFNγ-primed THP-1 ΔCASP1 cells transfected with soluble Tg antigen (STAg) that was untreated (UT) or treated as indicated. Area under the curve (AUC) from real-time assay plotted as mean ± SEM from n = 3 independent experiments.

C  Immunoblots from IFNγ-primed THP-1 ΔCASP1 or THP-1 ΔASC cells transfected with non-targeting control (CTRL), GBP1, AIM2, CASP1, or CASP8 siRNA and then untreated (−) or transfected with STAg for 12 h. Images are representative of n = 2 independent experiments.

D  AnnV-Glo assay from naïve or IFNγ-primed THP-1 ΔCASP1 cells transfected with non-targeting control (CTRL), GBP1, AIM2, CASP1, or CASP8 siRNA and then untreated (UT) or transfected with STAg as indicated for 18 h. Area under the curve (AUC) from real-time assay plotted as mean ± SEM from n = 3 independent experiments. Schematic on right shows an overview of pathways leading to caspase-8 activation.

E  Propidium iodide (PI) uptake assay from MDMs transfected with non-targeting control (CTRL) or GBP1 siRNA and then left untreated (UT), treated with LPS or transfection reagent (Lipofectamine) only, or transfected with LPS for 4 h. Area under the curve (AUC) from real-time assays plotted as mean ± SEM from n = 4 independent experiments. Schematic on right shows an overview of pathways leading to caspase-4 activation.

Data information: *P ≤ 0.05; ****P ≤ 0.0001 from two-way ANOVA following adjustment for multiple comparisons; ns, not significant.
Source data are available online for this figure.

microbial vacuoles has been identified previously (Finethy et al, 2017; Lindenberg et al, 2017; Santos et al, 2018). The correlation of microbe-targeting and cell death in our studies mirrored similar findings on mouse and human GBPs that target bacteria and restrict their replication in a GTPase- and isoprenylation-dependent manner (Kim et al, 2011; Finethy et al, 2015; Li et al, 2017; Piro et al, 2017;

Wandel et al, 2017). Interestingly, the recruitment of GBP1 to Tg vacuoles is cell-type specific, as it was shown to target PVs in mesenchymal stem cells (Qin et al, 2017) but not in A549 lung epithelial cells (Johnston et al, 2016), which correlates with the ability of GBP1 to induce cell death upon targeting Tg vacuoles. Similarly, murine Gbps do not target Chlamydia muridarum; however,

they are required for pyroptosis (Finethy *et al*, 2015). The exact mechanism by which GBPs interact with pathogen vacuoles and lead to their opening has been subject of previous studies (Yamamoto *et al*, 2012). However, no definitive mechanism has been identified. Our work provides additional evidence that GBP1 membrane interaction depends on its lipidation and GTPase activity, and suggest these properties contribute to vacuole opening. Further studies are needed to unravel factors that promote the loading of GBP1 onto microbial vacuoles and how GBPs restrict intracellular pathogens.

*Tg* induced AIM2-, ASC-, and caspase-8-dependent cell death in both primary MDMs and THP-1 cells. We found that *Tg*-infected macrophages formed ASC specks that contained active caspase-8. The sustained expression of AIM2 in *Tg*-infected macrophages and the loss of NLRP3 and caspase-1 might suppress the newly described STING-NLRP3-dependent cell death pathway (Gaidt *et al*, 2017). In this alternative DNA-sensing pathway, STING causes lysosomal damage-induced cell death and NLRP3 activation. Indeed, transfection of *Tg* DNA in ΔCASP1 cells triggered AIM2/ASC/caspase-8-dependent apoptosis, suggesting that loss of caspase-1 alone was sufficient to switch the cell death mechanism. The role of other cytosolic DNA-sensors in detecting *Tg* also needs be explored in the future (Roers *et al*, 2016). Similar to our findings on reduced NLRP3 activation in *Tg*-infected macrophages, *Tg* can also suppress NLRP3 in neutrophils (Lima *et al*, 2018). Rat and mouse NLRP1B trigger *Tg*-induced pyroptosis (Cirelli *et al*, 2014; Ewald *et al*, 2014; Gorfu *et al*, 2014) and human NLRP1 is implicated in the process in monocytes (Witola *et al*, 2011; Gov *et al*, 2017). Murine NLRP1B activation requires the liberation of its C-terminal caspase-1 activating region after undergoing self-cleavage and selective proteasomal degradation of the N-terminal fragment (Chui *et al*, 2019; Sandstrom *et al*, 2019). It is plausible that unknown *Tg* effectors fail to target monocyte NLPR1 and caspase-1. We hypothesize that *Tg* causes the total depletion of caspase-1, NLRP1, NLRP3, and NLRC4 through one or more uncharacterized *Tg* effector proteins in macrophages. Why the *Tg* effectors GRA35, 42, and 43, which are involved in inflammasome activation in rat macrophages (Wang *et al*, 2018), fail to do so in human macrophages needs to be addressed in the future.

STm activates the NAIP-NLRC4 and/or NLRP3 pathways in mouse (Franchi *et al*, 2006; Miao *et al*, 2006; Broz *et al*, 2010) and human macrophages (Kortmann *et al*, 2015; Reyes Ruiz *et al*, 2017). Our studies pinpoint a role for GBP1 in the enhancement of pyroptosis by IFNγ stimulation, trafficking, and activation of caspase-4 and confirm a previously described downstream role for GBP5 (Shenoy *et al*, 2012). GBPs can contribute to cytokine processing and/or pyroptosis by assisting in the activation of inflammasomes (Shenoy *et al*, 2012; Pilla *et al*, 2014; Feeley *et al*, 2017; Lagrange *et al*, 2018; Santos *et al*, 2018) or the release of microbial ligands by promoting microbial/vacuolar lysis (Meunier *et al*, 2014, 2015; Man *et al*, 2015; Balakrishnan *et al*, 2018; Liu *et al*, 2018). Importantly, both GBP1 and caspase-4 target STm, and caspase-4 targeting was GBP1-dependent. We hypothesize that GBP1 facilitates the opening of the bacterial vacuole and serves an essential role in exposing LPS for detection by caspase-4 (Shi *et al*, 2014). STm Δ*sifA*, which forms unstable vacuoles and enters the cytosol, is occasionally targeted by caspase-11 in murine macrophages (Thurston *et al*, 2016). In contrast, human caspase-4 can be activated in naïve macrophages infected with

*Francisella* (Lagrange *et al*, 2018) or enteropathogenic *E. coli* (Goddard *et al*, 2019), and the absence of a major role for caspase-4 in naïve macrophages infected with STm needs to be investigated further in the future.

We found that the role of GBP1 in cell death could be bypassed by transfecting molecules from *Tg* or STm, but not during natural infection, pointing to its role in liberating microbial ligands. Importantly, we have discovered microbe-specific activation of divergent host-cell death types downstream of GBPs. On a systemic level, depletion of macrophages by apoptosis and pyroptosis has advantages for both host and pathogen depending on the point-of-view. From the pathogen side, depletion of cells that constitute a major threat works to their advantage. On the host side, depletion of infected host cells reduces the number of infectious pathogens and limits the infection to one hot spot as infected macrophages no longer travel to other sites. Further, apoptosis may trigger a different adaptive immune response (Campisi *et al*, 2014), as compared to the exuberant inflammation triggered by pyroptosis, and contribute to the clearance of pathogens trapped in apoptotic cells by efferocytosis (Campisi *et al*, 2014). In summary, we establish that GBP1 targets microbial vacuoles and promotes different host cell death against diverse infectious agents. Our study expands the understanding of how IFNγ influences host cells during infection by two important pathogens.

## Materials and Methods

### Cells and parasites

THP-1 (TIB-202, ATCC) were maintained in RPMI with GlutaMAX (Gibco) supplemented with 10% heat-inactivated FBS (Sigma), at 37°C in 5% $CO_2$ atmosphere. THP-1 cells were differentiated with 50 ng/ml phorbol 12-myristate 13-acetate (PMA, P1585, Sigma) for 3 days and then rested for 2 days by replacing the differentiation medium with complete medium without PMA. Cells were not used beyond passage 20. HEK293T and human foreskin fibroblasts (HFF) were maintained in DMEM with GlutaMAX (Gibco) supplemented with 10% FBS at 37°C in 5% $CO_2$ atmosphere. THP-1 ΔCASP1, ΔCASP4, ΔASC (Schmid-Burgk *et al*, 2015), and $GSDMD^{miR}$ (Eldridge *et al*, 2017) were cultured as above. An overview of all cell lines made/used in this study is provided in Appendix Table S1.

*Tg* expressing luciferase/eGFP (RH type I and Prugniaud (Pru) type II) or expressing tdTomato were maintained *in vitro* by serial passage on monolayers of HFF cells, cultured in DMEM with GlutaMAX supplemented with 10% FBS, at 37°C in 5% $CO_2$. All cell culture was performed without addition of antibiotics unless otherwise indicated. Cell lines were routinely tested for mycoplasma contamination by immunofluorescence, PCR, and agar test.

### Human macrophage isolation and culture

For isolating primary human macrophages, peripheral blood mononuclear cells were extracted from leukocyte cones from healthy donors (NHS) via Ficoll (17544202, GE Healthcare) density gradient centrifugation. $CD14^+$ Monocytes were extracted using

magnetic microbeads (130-050-201, MACS Miltenyi). Monocytes were counted and seeded for 1 week in RPMI containing 10% human AB serum (H4522, Sigma), GlutaMAX (Gibco), penicillin/streptomycin (Gibco), and 5 ng/ml hGM-CSF (130-093-864, Miltenyi). The medium was replaced after 2 and 5 days, to replenish the hGM-CSF.

### Cell treatments

Cells were stimulated for 16 h prior to infection in complete medium at 37°C with addition of 50 IU/ml human IFN$\gamma$ (285-IF, R&D Systems). Treatments to induce apoptosis were performed with 50 ng/ml TNF$\alpha$ (210-TA, R&D Systems) and 10 µg/ml cycloheximide (CHX, C7698, Sigma) or 0.5 µg/ml staurosporine (S4400, Sigma). The caspase inhibitor zVAD-fmk (25 µM, 60332, Cell Signaling Technologies) or qVD (20 µM, ab141421, Abcam) was used. To chemically induce pyroptosis, cells were either treated with 10 µM nigericin (N1495, Invitrogen) or transfected with 100 ng LPS *Sm* (IAX-100-011, Adipogen) per 60,000 cells using Lipofectamine 2000 (11668027, Invitrogen). Induction of GBP1 expression in the Doxycycline-inducible cells was performed with 200 ng/ml Doxycycline (D9891, Sigma), which is below the $LD_{50}$ (6.4 µg/ml) of Dox for *Tg* (Chang *et al*, 1990). To entirely block translation for studying protein stability in time course experiments, the cells were treated with 50 µg/ml CHX.

### *Toxoplasma gondii* infection

Parasites were routinely grown in HFFs and passaged the day before infection to obtain parasites with a high viability for infection. *Tg* tachyzoites were harvested from HFFs by scraping the cells and syringe lysing the cells through a 25G needle. Following syringe lysis, the *Tg* suspension was cleared of cell debris by centrifugation at $50 \times g$ for 5 min and the supernatant transferred into a new tube. Parasites were pelleted by centrifugation at $500 \times g$ for 7 min, washed with complete medium, centrifuged again, and finally re-suspended in fresh medium. Viable parasites were counted with trypan blue and used for infection at a multiplicity of infection (MOI) of three for most experiments or 1 for immunofluorescence imaging. Infection was synchronized by centrifugation at $500 \times g$ for 5 min. Two hours after infection, extracellular parasites were removed with three washes with warm PBS and then cells cultured in fresh medium.

### *Salmonella* Typhimurium infection

STm strain SL1344 was transformed with pFVP25.1, which contains rpsM promoter driving GFPmut3 (Valdivia & Falkow, 1996), maintained under Ampicillin selection (11593027, Gibco), and grown on LB + Ampicillin agar plates. STm SL1344 WT strain was maintained under streptomycin (11860038, Gibco) selection and grown on LB + Streptomycin agar plates. STm SL1344 $\Delta invA$ strain (kind gift from Jorge Galán, Galán *et al*, 1992) was maintained under streptomycin and kanamycin (11815032, Gibco) selection and grown on LB + Kanamycin agar plates. The day before infection bacteria from a single colony were inoculated into 9 ml LB and grown overnight at 37°C. The overnight culture was diluted 1:20 into LB + 300 mM NaCl (746398, Sigma) and grown

with shaking in a closed container until an $OD_{600}$ of 0.9. Bacteria were harvested by centrifugation at $1,000 \times g$ for 5 min and washed with serum-free cell culture medium twice and re-suspended in 1 ml medium without FCS. Cells were infected with STm at an MOI of 30, and infections were synchronized by centrifugation at $750 \times g$ for 10 min. Infected cells were washed 30 min post-infection three times with warm PBS (806552, Sigma) to remove extracellular bacteria, and fresh medium containing 100 µg/ml gentamicin (15750060, Gibco) was added for 1 h. Medium was then replaced with medium containing 10 µg/ml gentamicin and the infection continued for indicated times. Bacterial MOI used for infections were confirmed by plating on LB agar plates.

### AnnV/PI flow cytometry

For flow cytometry, PMA-differentiated $1 \times 10^6$ THP-1 cells were harvested using accutase (A6964, Sigma) and washed twice with warm PBS and once in AnnV binding buffer (10 mM HEPES pH 7.4, 140 mM NaCl, 2.5 mM $CaCl_2$). Cells were re-suspended in AnnV binding buffer containing 1:100 AnnV-PacBlue (A35122, Invitrogen) and stained at 4°C in the dark for 15 min and washed twice with AnnV binding buffer. Cells were analyzed on a LSR Fortessa (BD Biosciences), and data were processed using FlowJo version 10.3 (FlowJo, LLC).

### Live time-lapse imaging of apoptosis

To measure apoptosis and caspase activity, THP-1 cells were seeded in an ImageLock 96-well plate (#4379, Essen Biosciences) and differentiated as described above. Cells were then primed with 50 IU/ml IFN$\gamma$ overnight and infected with type I or type II *Tg*-expressing tdTomato as previously described. To measure caspase activity, NucView™ 488 caspase-3 substrate was added to the medium (RPMI without phenol red + 10% FCS) to a final concentration of 5 µM. Following infection, the cell culture plate was placed in an IncuCyte® ZOOM (Essen Biosciences) and phase-contrast and green fluorescence images recorded every 15 min over a time course of 18 h. Images were analyzed using the IncuCyte® ZOOM software (Essen Biosciences).

### LDH and XTT assay

To determine cell viability and cell membrane integrity using XTT (Cell proliferation kit II, Roche) and LDH assays (Pierce LDH Cytotoxicity Assay Kit, 88953, Thermo Scientific), 25,000 THP-1 cells were seeded per well in a 96-well plate and differentiated with PMA. For both assays, cells were treated and infected in RPMI without FCS and without phenol red (11835, Gibco) to not interfere with the colorimetric measurements. For LDH assay, 24 h p.i. cell culture supernatant was harvested and cleared by centrifugation at $2,000 \times g$ for 5 min, and 50 µl was used to perform LDH assay according to the manufacturer's instruction (absorption at 490 and 680 nm for correction). For XTT assay, the detection reagent was freshly prepared according to the manufacturer's instruction and added 24 h p.i. with *Tg*. Cells were then incubated at 37°C for 4 h, and absorption was measured at 475 and 660 nm for correction.

## AnnV glow assay

Apoptosis kinetics were analyzed using the RealTime-Glo™ Annexin V Apoptosis Assay (JA1001, Promega) according to the manufacturer's instruction. To do so, $0.06 \times 10^6$ cells were seeded per well of a white, tissue culture-treated 96-well plates, differentiated, pre-treated, and infected as described before. Simultaneously with infection, detection reagent was added. Luminescence was measured using a Fluostar Omega plate reader (BMG Labtech). No-cell, medium-only controls were used for background correction.

## Real-time propidium iodide (PI) uptake assays

To measure live kinetics of necrotic cell death, $0.06 \times 10^6$ cells were seeded per well of a black-wall, clear-bottom 96-well plate (Corning) and differentiated with PMA, pre-treated, and infected as described above. Just before the start of the assay, medium was replaced with RPMI without phenol red and supplemented with 5 μg/ml propidium iodide (PI, P3566, Invitrogen). Then, the plate was sealed with a clear, adhesive optical plate seal (Applied Biosystems) and placed in a plate reader (Fluostar Omega, BMG Labtech) pre-heated to 37°C. PI fluorescence was recorded with top optics every 15 min for times as indicated.

## Immunoblotting

For immunoblotting, $0.5 \times 10^6$ cells were seeded per well of a 48-well plate and differentiated with PMA, pre-treated, and infected as described above. At the end of treatments, cells were washed with ice-cold PBS and lysed for 5 min on ice in 50 μl RIPA buffer supplemented with protease inhibitors (Protease Inhibitor Cocktail set III, EDTA free, Merck) and phosphatase inhibitors (PhosSTOP, Roche). Lysates were cleared by centrifugation at full speed for 15 min at 4°C and transferred into a new tube. Cleared lysates (diluted 1:5) were used for BCA assay (Pierce BCA protein assay kit, 23225, Thermo Scientific) to determine protein concentration. Samples (10 μg of total protein) were run on 4–12% Bis-Tris gels (Novex, Invitrogen) in MOPS running buffer and transferred on Nitrocellulose membranes using iBlot transfer system (Invitrogen). Depending on the primary antibody used, the membranes were blocked with either 5% BSA (A2058, Sigma) in TBS-T or 10% dry-milk (M7409, Sigma) in TBS-T for at least 1 h at room temperature. Incubation with primary antibodies was performed at 4°C overnight (all antibodies used in this study can be found in Appendix Table S2). To develop the blots, the membranes were washed with TBS-T, probed with 1:5,000 diluted secondary antibodies in 5% BSA in TBS-T, and washed again. Finally, the membranes were incubated for 2 min with ECL (Immobilon Western, WBKLS0500, Millipore) and luminescence was recorded on a ChemiDoc MP imaging system (Bio-Rad).

To perform immunoblots on culture supernatants, cells were infected/treated in OptiMEM (1105802, Gibco) without serum. Proteins in the supernatants were precipitated with 4× volume ice cold acetone (V800023, Sigma) overnight at −20°C, and pelleted by centrifugation at full speed for 10 min at 4°C. Pellets were air-dried for 10 min and re-suspended in 40 μl 2× Laemmli loading dye and used for immunoblotting.

## Quantitative RT–PCR (qRT–PCR)

RNA was extracted from $0.25 \times 10^6$ cells using Trizol reagent (15596026, Invitrogen). GlycoBlue (5 μg/ml, AM9516, Invitrogen) was added during the isopropanol (190764, Sigma) precipitation step to enhance precipitation. RNA quality was measured on a Nanodrop 2000 Spectrophotometer (Thermo Scientific). RNA (1 μg) was reverse transcribed using the high-capacity cDNA synthesis kit (4368813, Applied Biosystems). qPCR used the PowerUP SYBR green (A25742, Applied Biosystems) kit, 20 ng cDNA in a 20-μl reaction, and primers (Appendix Table S3) at 1 μM final concentration on a QuantStudio 12K Flex Real-Time PCR System (Applied Biosystems). Primer specificity was ensured by designing primers to span exon–exon junctions whenever possible, and for each primer pair, a melt curve was recorded and compared to *in silico* predicted melt curve using uMelt (Dwight *et al*, 2011). Recorded $C_t$ values were normalized to the recorded $C_t$ of human *HPRT1* and data plotted as $\Delta C_t$ (Relative expression).

## siRNA transfection

Cells were transfected with siRNAs 2 days prior to infection; at the same time, the THP-1 differentiation medium was replaced with medium without PMA or MDM differentiation medium was replaced on day 5 after seeding. All siRNAs were used at a final concentration of 30 nM. To prepare the transfection mix, a 10× mix was prepared in OptiMEM containing the appropriate siRNA(s) and TransIT-X2 transfection reagent (MIR 600×, Mirus) in a 1:2 stoichiometry. As the GBPs exhibit high sequence similarity, we created a custom transfection panel using three different silencer select siRNAs (Ambion):

| Gene | siRNA 1 | Conc. | siRNA 2 | Conc. | siRNA 3 | Conc. | Final Conc. |
|------|---------|-------|---------|-------|---------|-------|-------------|
| *GBP1* | s5620 | 20 nM | s5621 | 5 nM | s5622 | 5 nM | 30 nM |
| *GBP2* | s5623 | 20 nM | s5624 | 5 nM | s5625 | 5 nM | 30 nM |
| *GBP3* | s5626 | 10 nM | s5627 | 10 nM | s5628 | 10 nM | 30 nM |
| *GBP4* | s41805 | 7.5 nM | s41806 | 15 nM | s41807 | 7.5 nM | 30 nM |
| *GBP5* | s41808 | 5 nM | s41809 | 12.5 nM | s41810 | 12.5 nM | 30 nM |

The appropriate negative control was *Silencer*™ Select Negative Control No. 1 siRNA (#4390843, Ambion).

All other siRNAs used in this study were OnTarget Plus smart pool siRNAs (Dharmacon):

| Gene | Cat. number | Gene | Cat. number |
|------|-------------|------|-------------|
| CASP1 | L-004401 | CASP9 | L-003309 |
| CASP2 | L-003465 | CASP10 | L-004402 |
| CASP3 | L-004307 | NLRP1 | L-004423 |
| CASP4 | L-004404 | NLRP3 | L-017367 |
| CASP5 | L-004405 | NLRC4 | L-004396 |
| CASP6 | L-004406 | AIM2 | L-011951 |
| CASP7 | L-004407 | GSDMD | L-016207 |
| CASP8 | L-003466 | Negative control | D-001810 |

## Creation of THP-1 ΔGBP1

### Plasmid construction for generation of GBP1-deficient THP-1 cells

Oligonucleotides used for creation of the cells are listed in Appendix Table S4. GBP1-deficient cells were generated with the pX330 plasmid CRISPR/Cas9 system. The insert fragment of GBP1 gRNA1 or GBP1 gRNA2 was generated by annealing oligonucleotides and cloned into the *Bbs*I site of the cloning vector containing U6 promoter to generate gRNA-expressing plasmids pGBP1_gRNA1 and pGBP1_gRNA2, respectively. The *Xho*I and *Sal*I insert from the pGBP1_gRNA2 vector was cloned in the *Xho*I site of the pGBP1_gRNA1 vector to generate pGBP1_gRNA1/2 plasmid. The *Kpn*I and *Mlu*I fragment from pGBP1_gRNA1/2 was cloned into the *Kpn*I and *Mlu*I sites of pEF6-hCas9-Puro vector (Ohshima *et al*, 2014).

### Generation of GBP1-deficient THP-1 cells by CRISPR/Cas9 genome editing

$1 \times 10^6$ THP-1 cells were electroporated with 10 μg of the pEF6-hCas9-Puro vector containing target gRNA1/2 using NEPA21 (Nepa Gene). 48 h post-electroporation, 3 μg/ml puromycin was added for 10 days to select for cells with a stable integration. Cells were plated in limiting dilution in 96-well plates to isolate single cell clones. To confirm complete target gene deficiency, the targeted locus was sequenced, and the GBP1 protein expression analyzed by immunoblotting.

## Creation of Dox-inducible GBP1 cell lines

### Cloning of the Doxycycline-inducible system

To reduce leaky expression in the absence of Dox and maintain high responsiveness to Dox, we combined two transactivator proteins in one control vector: the transcriptional activator *rtTA2*$^S$-*M2* that induces expression in the presence of Dox and the silencer *tTS* that suppresses expression in the absence of Dox. *rtTA2*$^S$-*M2* was amplified by PCR from pBSKΔB-*CAG-rtTA2*$^S$-*M2-IRES-tTSkid-IRES-Neo* [Addgene #62346, a gift from Hiroshi Ochiai (Ochiai *et al*, 2015)], *tTS* from *UbC-TRE-mSOD2-tTS-IRES-EGFP* [Addgene #12390, a gift from Zuoshang Xu (Xia *et al*, 2006)], *P2A* and *T2A* from pSpCas9(BB)-*2A-GFP* (PX458) [Addgene #48138, a gift from Feng Zhang (Ran *et al*, 2013)], and *bsr* from pHH3-*SP-lox-GFP-lox* [a gift from Anthony Holder (Knuepfer *et al*, 2017)

with 30 bp overlaps to each other using Q5 DNA-polymerase (M0491, NEB)]. pLSC-5 [Addgene #62889, a gift from Feng Zhang (Zetsche *et al*, 2015)] which has an EFsNS promoter was digested with *Eco*RI and *Kpn*I to remove the sgRNA cassette and re-ligated with repair oligonucleotides and then amplified the backbone with 30 bp overlaps to parts of our new insert without the *splitCas-9* ORF. A 6-fragment Gibson assembly with home-made Gibson master mix created the pTet-ctrl plasmid (with *rtTA2*$^S$-*M2-P2A-tTs-T2A-bsr* fusion gene under the control of EFsNS promoter in a vector backbone suitable for lentiviral transduction into mammalian cells). All oligonucleotide sequences used for cloning can be found in Appendix Table S4.

To create a response vector, we used pLSC5 without the sgRNA cassette and replaced the *splitCas-9* ORF with a Zeocin resistance gene (*shBle*) by Gibson assembly [*shBle* was amplified from pSpCas9 (BB)-*2A-GFP* (pX458)] to generate pLSC-5-*Zeocin*. The tet response element (TRE) consisting of a minimal CMV promoter downstream of a heptamerized tet-operator (Gossen & Bujard, 1992) from WT *HA-dynamin 1 pUHD10-3* [Addgene #34688, a gift from Sandra Schmid (Damke *et al*, 1995)] and pLSC-5-*Zeocin* were used for Gibson assembly to generate a plasmid (pLenti-Tet) suitable for lentiviral transduction into mammalian cells that contains a TRE promoter and the Zeocin resistance under control of an EFsNS promoter. Finally, a multiple cloning site was inserted in the *Bam*HI and *Eco*RI sites downstream of the TRE to make the vector more versatile.

### GBP1 plasmids

To create plasmids that express GBP1 under the control of Dox, RNA from IFNγ-treated THP-1s was extracted and cDNA was synthesized. The CDS of the *GBP1* mRNA was amplified with Q5 polymerase and the amplicon treated with Taq polymerase (M0273, NEB) to create A overhangs. Subsequently, the amplicon was cloned into pCR2.1®-TOPO TA vector using TOPO TA kit (451641, Invitrogen). Next, GBP1 mutants were created by site-directed mutagenesis, introducing single point mutations. Using the mutated or WT *GBP1*-containing vectors, the ORFs were amplified using oligonucleotides to create overhangs to pLenti-Tet vector. The target vector itself was digested using *Bam*HI. We then performed Gibson assembly of the digested backbone and the *GBP1* ORFs and confirmed successful cloning by sequencing. To create mCH-tagged versions, we additionally amplified *mCH* ORF with overlaps to the backbone and *GBP1* and included it in the Gibson assembly reactions.

### Lentiviral transduction of the Dox-inducible system

Lentiviral packaging used HEK293T cells transfected with required plasmids, pMD2.G (Addgene #12259, a gift from Didier Trono) and psPAX2 (Addgene #12260, a gift from Didier Trono) using Lipofectamine 2000 in serum-free DMEM. Medium was replaced 12 h post-transfection with fresh DMEM + 10% FBS to rest the cells. Medium was replaced again after 12 h with DMEM + 10% FBS containing 5 mM sodium butyrate (B5887, Sigma) to boost virus production and left for another 12 h. Medium was replaced with RPMI + 10% FBS and the cells left to produce lentiviral particles for 1 day. Virus-containing supernatant was filtered through a 0.43-μm syringe filter and supplemented with 8 μM polybrene (H9268, Sigma). The THP-1 ΔGBP1 target cells were then re-suspended in 500 μl of the virus-containing medium and "spinfected" for 30 min at 720 × *g* before being placed back in the incubator. One hour p.i. 1 ml complete

medium was added, and the cells left to rest. This infection procedure was repeated for a total of three times before selection with 15 μg/ml Blasticidin S (15205, Sigma). Once untransduced control cells had died in the selection medium, the newly created THP-1 ΔGBP1 + Tet cells were then transduced with the GBP1 ORF-containing response vectors (pLenti-Tet) following the same procedure and selection was performed using 200 μg/ml Zeocin (J67140, Alfa Aesar).

### Creation of YFP-CASP4-expressing cell lines

#### Cloning of YFP-CASP4 retroviral vectors
CASP4 and CASP4^C258S ORF was amplified by PCR (Pallett et al, 2017) from plasmids kindly gifted by Mitchel Pallett and cloned into pMX-CMV-YFP [CMV-driven expression of YFP derived from pEYFPC1 (Clontech; Eldridge et al, 2017) via sequence- and ligation-independent cloning (SLIC)]. CASP4^C258S mutation was generated by cloning two overlapping PCR fragments (by Mitchel Pallett) and was used as template for PCR to generate plasmid pMX-CMV-YFP-CASP4^C258S.

#### Retroviral transduction of caspase-4 into THP-1 cells
Caspase-4-encoding plasmids and the packaging plasmids pCMV-MMLV-pack (Shenoy et al, 2012) and pVSVg were transfected into HEK293T cells in serum-free DMEM using Lipofectamine 2000 at a ratio of 5:4:1, to produce retroviral particles for transduction. Medium was replaced 12 h post-transfection with fresh DMEM + 10% FBS to rest the cells. Medium was replaced again after 12 h with DMEM + 10% FBS containing 5 mM sodium butyrate to boost virus production and left for another 12 h. Medium was replaced with RPMI + 10% FBS and the cells left to produce lentiviral particles for 1 day. Virus-containing supernatant was filtered through a 0.43-μm syringe filter and supplemented with 8 μM polybrene. The THP-1 ΔGBP1 + Tet-mCH-GBP1 target cells were then re-suspended in 500 μl of the virus-containing medium and "spinfected" for 30 min at 720 × g before being placed back in the incubator. One hour p.i. 1 ml complete medium was added, and the cells left to rest. This infection procedure was repeated for a total of three times before selection with 1 μg/ml puromycin. Once untransduced control cells had died in the selection medium, the newly created cells were tested by immunoblotting and subsequently used for experiments.

### IL-1β and IL-18 ELISA

To measure the production and secretion of cytokines by infected THP-1s, the cell culture supernatant was harvested and cleared by centrifugation at 2,000 × g for 5 min. The cleared supernatants were then diluted in the buffer provided with the ELISA kits, and ELISA was performed according to the manufacturer's instruction. IL-1β ELISA kit was from Invitrogen (#88-7261, detection range 2–150 pg/ml), and IL-18 ELISA was from R&D Systems (#7620, detection range 12.5–1,000 pg/ml).

### Fixed immunofluorescence microscopy

For confocal imaging, $0.25 \times 10^6$ THP-1 cells were seeded on gelatin-coated (G1890, Sigma) coverslips in 24-well plates and differentiated with PMA. Following treatments and infection, cells were washed three times with warm PBS prior to fixation to remove any uninvaded pathogens and then fixed with 4% methanol-free formaldehyde (28906, Thermo Scientific) for 15 min at room temperature. Following fixation, cells were washed again with PBS and kept at 4°C overnight to remove any unreacted formaldehyde. Fixed specimens were permeabilized with PermQuench buffer (0.2% (w/v) BSA and 0.02% (w/v) saponin in PBS) for 30 min at room temperature and then stained with primary antibodies for 1 h at room temperature. After three washes with PBS, cells were incubated with the appropriated secondary antibody and 1 μg/ml Hoechst 33342 (H3570, Invitrogen) diluted in PermQuench buffer for 1 h at room temperature. Cells were washed with PBS five times and mounted using 5 μl ProLong™ Gold Antifade Mountant (P36930, Invitrogen).

For imaging of caspase-8 recruitment to ASC specks, FITC-IETD-fmk (88-7005-42, Invitrogen) was added to the culture medium 1 h prior to fixation. Cells were then fixed and stained as detailed above.

Specificity of the home-made GBP1-antibody was validated by staining IFNγ-primed THP-1 ΔGBP1 + Tet-GBP1 cells (express all GBPs but GBP1) that were additionally treated with Doxycycline as a positive control (Appendix Fig S2).

Specimens were imaged on a Leica SP5-inverted confocal microscope using 100× magnification and analyzed using LAS-AF software. Images were further formatted using FIJI software.

### Quantification of protein recruitment to pathogen vacuoles and ASC speck formation

Cells seeded as above were infected with Tg MOI = 1 and fixed 6 h p.i. or STm-GFP MOI = 30 and fixed 2 h p.i. Nuclei were stained with Hoechst 33342 and coverslips mounted as described above. Images were acquired using a Ti-E Nikon microscope equipped with LED-illumination and an Orca-Flash4 camera using a 60× magnification. All intracellular vacuoles of 100 fields of view were automatically counted based on whether they show recruitment of GBP1, caspase-4, both, or no recruitment using HRMAn high-content image analysis (Fisch et al, 2019). Further, the analysis pipeline was used to measure the fluorescence intensity of GBP1 on STm vacuoles using the radial intensity measurement implemented in HRMAn. The Pearson's correlation coefficient for colocalization analysis was also computed on cropped STm vacuoles, that have been classified as decorated with both proteins by HRMAn, using Fiji. The experiment was repeated independently three times.

For quantification of ASC speck formation and caspase-8 recruitment, 100 Tg-infected cells were manually counted per condition using a Ti-E Nikon microscope equipped with LED-illumination using a 100× magnification based on whether they contain an ASC speck and whether caspase-8 was recruited to this speck. The experiment was repeated independently three times.

### STAg preparation, pre-digestion, and transfection

STAg was prepared as described previously (Gazzinelli et al, 1991). In brief, parasites from a confluent T25 flask were lysed by freezing at −80°C and thawing at 37°C followed by four 20-s rounds of sonication. The resulting homogenate was centrifuged at 590 × g for 10 min, and supernatant collected. To deplete STAg of specific components, it was pre-treated with either 5 U/rxn of Turbo DNase (AM2238, Invitrogen) to digest DNA, 10 μg/ml RNase A (EN0531, Thermo Scientific) to digest remaining RNA, trypsin (90057, Thermo

Scientific) to digest proteins [protease to protein ratio 1:20 (w/w)], or with Protein Deglycosylation Mix II (P6044, NEB) to get rid of any glycosylation overnight. Untreated or pre-treated STAg was then transfected into THP-1s using Lipofectamine 2000 (1 μl STAg per well of a 96-well plate).

## Data handling and statistics

Graphs were plotted using Prism 8.0.2 (GraphPad Inc.) and presented as means of $N = 3$ experiments (with usually three technical repeats within each experiment) with error bars representing SEM, if not stated otherwise. Data analysis used one-way ANOVA, two-way ANOVA, nested $t$-test, or unpaired Student's $t$-test as indicated in the figure legends. Benjamini, Krieger, and Yekutieli false-discovery rate ($Q = 5\%$)-based correction for multiple comparisons as implemented in Prism was used when making more than three comparisons.

Expanded View for this article is available online.

## Acknowledgements

We would like to thank Monique Bunyan for preparing STAg, Asha Patel for help with flow cytometry, Max Gutierrez and Daniel Greenwood for sourcing and advising MDM preparation, Anna Coussens and Nashied Peton for advice on MDM differentiation and transfection, Julia Sanchez-Garrido for help in optimizing immunoblots and advice on siRNAs, the Crick High-throughput screening (HTS) STP for help in performing the Incucyte time course experiments, the Crick Genomics and Equipment Park STP for performing sequencing and DNA minipreps for cloning, and Jörn Coers for critical reading of the article. We thank all members of the Frickel and the Shenoy laboratories for productive discussion and Crick Core facilities for assistance in the project. This work was supported by the Francis Crick Institute, which receives its core funding from Cancer Research UK (FC001076), the UK Medical Research Council (FC001076), and the Wellcome Trust (FC001076). EMF was supported by a Wellcome Trust Career Development Fellowship (091664/B/10/Z). DF was supported by a Boehringer Ingelheim Fonds PhD fellowship. ARS would like to acknowledge support from the MRC (MR/P022138/1) and Wellcome Trust (108246/Z/15/Z). MY was supported by the Research Program on Emerging and Re-emerging Infectious Diseases (JP18fk0108047) and Japanese Initiative for Progress of Research on Infectious Diseases for global Epidemic (JP18fk0108046) from Agency for Medical Research and Development (AMED). HB was supported by Grant-in-Aid for Scientific Research on Innovative Areas (17K15677) from Ministry of Education, Culture, Sports, Science, and Technology.

## Author contribution

DF, ARS, and E-MF conceived the idea for the study; DF performed experiments; and BC provided essential experimental protocols. HB, MY, and VH provided essential reagents. DF, ARS, and E-MF analyzed and interpreted the data and wrote the article. All authors revised the article.

## Conflict of interest

The authors declare that they have no conflict of interest.

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
