## [Review Process File · The EMBO Journal]

Human GBP1 is a microbe-specific gatekeeper of macrophage apoptosis and pyroptosis

Daniel Fisch, Hironori Bando, Barbara Clough, Veit Hornung, Masahiro Yamamoto, Avinash R. Shenoy, Eva-Maria Frickel

Review timeline:	Submission date:	4th Dec 2018
	Editorial Decision:	15th Jan 2019
	Revision received:	28th Mar 2019
	Editorial Decision:	7th May 2019
	Revision received:	10th May 2019
	Accepted:	13th May 2019

Editor: Karin Dumstrei

Transaction Report:

1st Editorial Decision

15th Jan 2019

Thank you for submitting your manuscript to The EMBO Journal. I am sorry for the slight delay in getting back to you with a decision, but I have now received the comments from the two referees on the manuscript.

As you can see from the comments below, the referees find the analysis interesting but they also bring up a number of points that would have to be resolved in order to consider publication here. Looking at the concerns raised I anticipate that you should be able to resolve them in a good way. Given the concerns raised I would therefore like to invite you submit a revised version.

Let me know if we need to discuss any of specific points raised in further detail.

I should add that it is EMBO Journal policy to allow only a single round of revision, and that it is therefore important to address the major concerns at this stage.

REFeree REPORTS:

Referee #1:

General summary and opinion about the principle significance of the study, its questions and findings

In this well written paper the authors investigate how GBP1 influences cell death pathways drive by Toxoplasma and Salmonella. They claim that GBP1 acts as a gatekeeper of cell-death pathways, which respond specifically to infecting microbes and suggest that their findings expand the immune roles of human GBPs in regulating not only pyroptosis, but also apoptosis. The paper contains a large amount of high quality experimental data which supports many, but not all of their claims. Overall the Toxoplasma data is supportive of the authors' hypothesis, but the Salmonella data is

more problematic. This latter data set shows GBP1 recruitment to Salmonella, but it is difficult to interpret whether this plays a significant role in Salmonella induced cell death enhancement by IFN because the levels of basal cell death (10-15%) are very low and the IFN enhancement small (to 15-20%).

Specific major concerns essential to be addressed to support the conclusions

The hypothesis on Tg and the GBP1-dependent IFN enhancement is broadly supported by the data presented. The Salmonella data is not compelling, looks biologically insignificant and needs major revision with additional experiments. The data does nicely show that GBP1 is recruited to Salmonella during infection and suggests that GBP1 facilitates IFN-Salmonella responses but it is unclear how important this is in cell death because the overall rate of cell death and the increases induced by IFN are so small.

1. What MOI were used for this work? The basal levels of Salmonella cell death in THP1 (5%) and MDM (15%) very low and not consistent with other publications which show much higher levels of cell death (depending upon MOI). The IFN increase in cell death also small. It is therefore currently unclear whether the changes in cell death induced by IFN are biologically relevant. Low MOIs produce low levels of cell death, whereas increasing MOIs of 10 and above induce significant cell death in bacteria in the log phase of bacterial growth. Assuming the authors grew their bacteria to log phase (from the methods this is possible) then high levels of rapid cell death should occur within an hour. The LDH release is low and the PI uptake low when compared to the Tg induced comparable responses (the axes are different). Experiments with different MOIs should be performed plus/minus IFN and the GBP1 dependency considered.

2. What happens when the authors use WT Salmonella as opposed to genetically modified Salmonella? Why did the authors not use a strain of Salmonella where by the genetic modification is on the chromosome so antibiotic selection not required? This might explain the differences seen in cell death in comparison to other published studies and WT Salmonella should also be used for experiments in this study.

3. Fig 3D The caspase 1 and caspase 4 individual data represent very small differences unlike the combined Caspase 1 Caspase 4 double KO. How were the statistical analysis done because the data is not convincing?

4. The data in fig 3H is also not convincing that there are any differences at all between UT and IFN treatment

Minor concerns that should be addressed

1. How does IFN reveal a caspase 4 cell death given log phase Salmonella are not thought to kill cells via caspase 4 only stationary phase bacteria are supposed to do this (in mouse cells anyway although this may not be the case in human cells)? Presumably the hypothesis is IFN induces caspase 4 expression?

2. Fig 2 C the cleaved caspase 8 not compelling and a better blot should be used

3. S3: IL1 data has wide error bars: do the authors have an explanation as to why this is?

4. Broz & Dixit, 2016 is not reference for cell death induced by Salmonella so the primary sources rather than a review should be cited here

Referee #2:

The article by Fisch et al. shows that human GBP1 drives *Toxoplasma gondii* mediated cell death by targeting the parasite-containing vacuole (PCV) during infection. The authors propose that GBP1 recruitment to the PCV promotes AIM2-dependent but GSDMD-independent cell death via ASC and caspase-8 mediated apoptosis. Conceptually this is poorly supported by the experiments due to the authors not carefully examining the mechanism(s) of cell death in the absence of pyroptosis mediators during *T. gondii* infection. The microscopy data with tagged GBP1 mutants is nicely executed and convincing. The authors also show that human GBP1 mediates release of Salmonella ligands to activate pyroptosis, which is entirely expected based on previous studies on murine GBPs and Salmonella, but important to confirm in the context of human GBPs. Overall, the findings with *Toxoplasma* infection are very interesting but multiple additional experiments should be performed to better explain the mechanism of cell death that is dependent upon GBP1, AIM2, and ASC. Additional experiments are also needed to provide insights on how GBP1 recruitment to PCV facilitates the release of DNA from *T. gondii*.

Major Comments:

1. The authors use LDH release and PI uptake as measurements for cell death and XTT for cell viability but should also provide microscopy images showing the cell morphology during cell death. The morphology is especially important to consider because of the unique mechanism of cell death proposed.
2. In addition to ASC/Casp-1, ASC/Caspase-8 interaction and cross-talk has also been observed before (Man et al. J. Immunol. 2013, others). Is caspase-8 recruited to ASC specks in caspase-1 deficient cells? Does specific inhibition of caspase-8 rescue cells from death (in the presence or absence of caspase-1/4)?
3. Pro- and cleaved caspase-1 blots should be shown for unprimed, IFN γ primed, and GBP1-deficient (unprimed and primed) during infection. Just because cells die in the absence of caspase-1, the role for canonical pyroptosis in caspase-1 sufficient cells should be determined in the context of GBP1 presence and absence. The authors should also explain why pro-caspase-1 is decreasing over time during infection (could be as simple as caspase-1 cleavage through canonical AIM2 inflammasome) in Figure 2H and Fig. S3H. Annexin V is also known to label cells undergoing pyroptosis, suggesting it is not a specific marker for apoptosis (Vasconcelos et al. Cell Death and Differentiation 2018).
4. The cell-type specificity in the recruitment of GBP1 to PCV is interesting. Are there any differences in the replication of *T. gondii* or the structure of PCV between different cell types? Does the cell-type specific recruitment of GBP1 also leads to differences in the mode of cell death in response to *T. gondii* infection?
5. The authors propose GBP1-mediated release of DNA from the parasite as the preceding event that leads to AIM2-, ASC- and caspase8- dependent cell death. The authors should provide some insights regarding the mechanisms by which GBP1 recruitment to PCV leads to DNA release for sensing by AIM2.

Minor Comments:

1. Discuss why NLRP1 inflammasome is not involved in mediating cell death to *Toxoplasma* in your infection model, as it has been previously described.
2. The specificity of the home-made antibody against GBP1 looks good by immunoblot but is not validated for immunofluorescence (Figure 3A, first panels in magenta?). GBP antibodies are notoriously cross-reactive.

1st Revision - authors' response

28th Mar 2019

Fisch et al – authors' response to reviewer' comments.

As several new figure panels were added based on experiments that addressed reviewers' concerns and re-formatting as per EMBO J guidelines, we have split two main figures (Figures 2 and 3) and renamed and reorganised some Supplementary Data. Their positions in the text are indicated below.

Referee #1:

General summary and opinion about the principle significance of the study, its questions and findings

In this well written paper the authors investigate how GBP1 influences cell death pathways driven by *Toxoplasma* and *Salmonella*. They claim that GBP1 acts as a gatekeeper of cell-death pathways, which respond specifically to infecting microbes and suggest that their findings expand the immune roles of human GBPs in regulating not only pyroptosis, but also apoptosis. The paper contains a large amount of high-quality experimental data which supports many, but not all of their claims. Overall the *Toxoplasma* data is supportive of the authors' hypothesis, but the *Salmonella* data is

more problematic. This latter data set shows GBP1 recruitment to *Salmonella*, but it is difficult to interpret whether this plays a significant role in *Salmonella* induced cell death enhancement by IFN γ because the levels of basal cell death (10-15%) are very low and the IFN γ enhancement small (to 15-20%).

We thank the reviewer for their overall positive comments.

The IFN γ signalling pathway is important in humans as seen from loss-of-function mutations. The enhancement in cell death upon IFN γ treatment is ~2-5 fold over untreated cells (and not 15-20%). GBPs are not integral to pyroptosis, and we and other groups have previously identified GBP5 in the enhancement of inflammasome activation by pathogens (Shenoy et al, 2012) and OMVs containing LPS (Santos et al, 2018). The reviewer will note that we have carefully worded our conclusions, and these are fully supported by previous and substantial new data (see additional comments below): we have said that GBP1 is not required for pyroptosis in STm-infected naïve cells; however, we not only identify that mechanism of enhancement of pyroptosis in IFN γ -treated cells (via caspase-4), but also show that caspase-4 activation by STm is fully reliant on GBP1. This is important new insight in the process and of broad interest to the field.

Specific major concerns essential to be addressed to support the conclusions

The hypothesis on Tg and the GBP1-dependent IFN γ enhancement is broadly supported by the data presented. The *Salmonella* data is not compelling, looks biologically insignificant and needs major revision with additional experiments. The data does nicely show that GBP1 is recruited to *Salmonella* during infection and suggests that GBP1 facilitates IFN γ -*Salmonella* responses but it is unclear how important this is in cell death because the overall rate of cell death and the increases induced by IFN γ are so small.

*We respectfully disagree that the 2-3-fold increase with IFN γ -priming is biologically insignificant. The identification of GBP1-dependent initiation of a distinct cell death pathway is an important finding. The reviewer will appreciate that given the differences with the mouse, addressing the in vivo role of GBP1 will require significant additional work e.g. humanised mouse models etc., which is beyond the scope of this revision. We believe that understanding the mechanisms of pyroptosis induced by an important human pathogen that IFN γ protects against is biologically extremely meaningful. However, we have taken the reviewer's comments onboard and further investigated how GBP1-targeting to STm influences cell death. New data (see **Figures 5I-J** and **EV4K**) which show that GBP1 is required for the targeting of caspase-4 to STm. Caspase-4 is completely excluded from STm in the absence of GBP1. This explains the enhanced pyroptosis in IFN γ -treated cells and the dependence on GBP1.*

1. What MOI were used for this work? The basal levels of *Salmonella* cell death in THP-1 (5%) and MDM (15%) very low and not consistent with other publications which show much higher levels of cell death (depending upon MOI).

*Indeed, MOIs used were not indicated in the Methods and we apologise sincerely for this oversight. MOI of 30 were used throughout and we have performed additional experiments that indicate that the effect of IFN γ is observed at multiple MOIs tested (new data in **Figure 5A**).*

The IFN γ increase in cell death also small. It is therefore currently unclear whether the changes in cell death induced by IFN γ are biologically relevant. Low MOIs produce low levels of cell death, whereas increasing MOIs of 10 and above induce significant cell death in bacteria in the log phase of bacterial growth. Assuming the authors grew their bacteria to log phase (from the methods this is possible) then high levels of rapid cell death should occur within an hour.

*See point above for response to IFN γ . We have used log-phase bacteria (as described in Methods along with MOI, lines 1016-1034). The reviewer is correct that in murine BMDMs log-phase STm activate NLRC4 (and only this pathway) rapidly and to very high levels. But this is not the case in human macrophages e.g. see Shenoy et al, 2012, Reyes Ruiz et al, 2017. In addition, we use a SPI-1 deletion strain (*Δ invA*) and show that STm-induced cell death requires this T3SS (**Figure 5A**).*

On a related note, we have recently shown that unlike in mouse BMDMs, pathogenic E. coli activate a caspase-4-dependent atypical pathway via NLRP3-caspase-1 in human MDMs that requires the bacterial T3SS and effectors (Goddard et al, 2019, in press). Altogether, there is an urgent need to investigate these pathways in human macrophages.

The LDH release is low and the PI uptake low when compared to the Tg induced comparable responses (the axes are different).

We do not compare cell-death caused by Tg (apoptosis) to that by STm (pyroptosis) due to MOI and other biological differences.

With regards to lower cell death than some other studies: (1) firstly, we have used primary MDMs which may show variability across geographical locations and previous work. (2) Our previous work (Shenoy et al, 2012) and that of others (Reyes Ruiz et al, 2017) shows that STm-infection in human macrophages does not only engage the NLRC4 inflammasome even though the direct delivery of STm T3SS rod/needle proteins in PAM3CSK4-primed MDMs or infection of rod/needle expressing Listeria in PAM3CSK4-primed MDMs leads to partially NAIP/NLRC4-dependent inflammasome activation. Note that Shenoy et al used LPS-primed cells (which increases NLRP3 expression and enhances pyroptosis as compared to non-primed cells) and Reyes Ruiz et al used experimental systems free of LPS and therefore could not have activated caspase-4. We used STm (or Tg) in TLR-unprimed MDMs and THP-1 cells to avoid non-physiological TLR-ligands. This explains the slightly lower levels of cell death we have observed. (3) As the concentration of PMA used for differentiation markedly affects STm-mediated pyroptosis (Starr et al, 2018), we used low levels (50 ng/ml) PMA for 3 days and then rested cells for 2 days to avoid nonspecific effects.

Altogether, the slightly lower absolute levels of cell death are due to above factors and do not affect our overall inferences. Moreover, the exciting findings in our study are the elucidation of the GBP1 and caspase-4 targeting of STm in IFN γ -activated cells.

Experiments with different MOIs should be performed plus/minus IFN γ and the GBP1 dependency considered.

*We thank the reviewer for this suggestion and have shown these data in **Figure 5A**. The dependence of enhanced pyroptosis on GBP1/IFN γ is seen at all MOIs tested.*

2. What happens when the authors use WT Salmonella as opposed to genetically modified Salmonella? Why did the authors not use a strain of Salmonella where by the genetic modification is on the chromosome so antibiotic selection not required? This might explain the differences seen in cell death in comparison to other published studies and WT Salmonella should also be used for experiments in this study.

*GFP-expressing STm were used to allow immunofluorescence studies. Such strains have been extensively used in the field of Salmonella and they do not affect most pathogenic processes (for instance, Yu et al, 2002; Catron et al, 2004; Henry et al, 2005, 2006; Antunes et al, 2012 have used pFVP25.1 for GFP expression in STm). However, we do appreciate the reviewer's point and have performed experiments with GFP-expressing and parental STm strains and found no difference in pyroptosis induction in naïve and IFN γ -primed macrophages – see new data in **Figure EV4E**. Experiments that used GFP bacteria are now indicated in Figure Legends.*

3. Fig 3D (now **Figure 5C**), the caspase 1 and caspase 4 individual data represent very small differences unlike the combined Caspase 1 Caspase 4 double KO. How were the statistical analysis done because the data is not convincing?

*The reviewer is correct that caspase-1 and caspase-4 individually contribute partially in IFN γ -primed cells (schematic in **Figure 5D**). However, we disagree with the reviewer's assessment and as we have had help from Statistical Advisory Service at Imperial College on similar studies (see Sanchez-Garrido et al, 2018), we are confident of our analyses. Values from n = 4 biologically independent experiments were analysed by two-way ANOVA (siRNA and IFN γ -priming as factors), and the Benjamini, Krieger and Yekutieli false-discovery rate ($Q = 5\%$) based corrected for multiple comparisons. Corrected $P < 0.05$ are indicated.*

To help with the inferences, we depict data here after normalising to STm-infected unprimed cells, which reveal fold-changes in cell death. IFN γ treatment enhanced cell death by ~80% as compared to unprimed cells. Silencing of caspase-1 or GSDMD alone results in ~70% reduction in cell death in naïve cells, and caspase-1+caspase-4 or GSDMD ~65% reduction in IFN γ -primed cells. Silencing either caspase-1 or caspase-4 alone in IFN γ -primed cells leads to a reduction back to levels observed in unprimed cells. However, we prefer to show 'raw' data (as in the Figures) than normalised data.

4. The data in fig 3H (now **Figure 5G**) is also not convincing that there are any differences at all between UT and IFN γ treatment

The reviewer is not right – there is a small but statistically significant difference between unprimed and IFN γ -primed THP-1s and MDMs ($P < 0.001$). Note that low levels of GBP1 expression can be detected in unprimed THP-1s and MDMs (new **Figure EV1B**), which is why some GBP1 recruitment to STm (and Tg; **Figure 4B**) can be observed even in the absence of IFN γ -priming. Importantly, the increased abundance of GBP1 leads to two things: (i) more bacteria (~2-fold) become positive for GBP1 (**Figure 5G**), which correlates with increased pyroptosis; (ii) there is more GBP1 (as quantified from staining intensity) on individual SCVs (new **Figure 5H**). Low-basal expression of GBP1 is consistent with a recent study in mouse macrophages showing that type I interferons maintain constitutive Gbp expression for rapid pyroptosis during Legionella infection (Liu et al, 2018).

Minor concerns that should be addressed

1. How does IFN γ reveal a caspase 4 cell death given log phase Salmonella are not thought to kill cells via caspase 4 only stationary phase bacteria are supposed to do this (in mouse cells anyway although this may not be the case in human cells)?

The reviewer is right that log-phase STm are detected by NLRC4 in murine BMDMs. As outlined above, this is not the case in human macrophages. In addition, we have a manuscript in press that describes rapid and atypical activation of caspase-4-NLRP3 by enteropathogenic E.coli (EPEC) that is strictly dependent on its T3SS system and secretion of the effector Tir (Goddard et al, 2019, in press) and is unique to human macrophages (Rathinam et al, 2012; Vanaja et al, 2016).

Presumably the hypothesis is IFN γ induces caspase 4 expression?

Unlike mouse caspase-11, human caspase-4 is not regulated by IFN γ transcriptionally. We investigated this further and our new data in **Figures 5I-J** and **EV4K** reveal GBP1-dependent recruitment of caspase-4 to STm. No caspase-4 was present on STm that were not already coated with GBP1. Together with data on cell death, these findings strongly support an IFN γ -GBP1-caspase-4 pathway during STm infection.

2. Fig 2C (still **Figure 2C**) the cleaved caspase 8 not compelling and a better blot should be used.

We have repeated these experiments with two additional MDM donors, and a new immunoblot showing active caspase-8 is now shown in **Figure 2C**.

3. S3 (now **Figure EV3C**): IL1 data has wide error bars: do the authors have an explanation as to why this is?

*The most likely explanation we have is that these experiments were performed with different batches of recombinant human IFN γ to stimulate the cells and these IFN γ batches might have varied in activities. We have now indicated matching values with connecting lines (see new **Figure EV3C**).*

4. Broz & Dixit, 2016 is not reference for cell death induced by Salmonella so the primary sources rather than a review should be cited here.

We agree and apologise for citing a review on this topic that is central to the second part of our manuscript. The two original papers on Salmonella-detection by NLRC4 inflammasomes are now cited (Miao et al, 2006; Franchi et al, 2006), along with the primary paper showing the redundant role of NLRC4 and NLRP3 on STm grown to stationary-phase (Broz et al, 2010).

Referee #2:

The article by Fisch et al. shows that human GBP1 drives Toxoplasma gondii mediated cell death by targeting the parasite-containing vacuole (PCV) during infection. The authors propose that GBP1 recruitment to the PCV promotes AIM2-dependent but GSDMD-independent cell death via ASC and caspase-8 mediated apoptosis. Conceptually this is poorly supported by the experiments due to the authors not carefully examining the mechanism(s) of cell death in the absence of pyroptosis mediators during T. gondii infection. The microscopy data with tagged GBP1 mutants is nicely executed and convincing. The authors also show that human GBP1 mediates release of Salmonella ligands to activate pyroptosis, which is entirely expected based on previous studies on murine GBPs and Salmonella, but important to confirm in the context of human GBPs. Overall, the findings with Toxoplasma infection are very interesting but multiple additional experiments should be performed to better explain the mechanism of cell death that is dependent upon GBP1, AIM2, and ASC. Additional experiments are also needed to provide insights on how GBP1 recruitment to PCV facilitates the release of DNA from T. gondii.

We thank the reviewer for their comments and have performed several additional experiments to support our conclusions.

Major Comments:

1. The authors use LDH release and PI uptake as measurements for cell death and XTT for cell viability but should also provide microscopy images showing the cell morphology during cell death. The morphology is especially important to consider because of the unique mechanism of cell death proposed.

*This is an important point and our new data further support apoptotic cell death in IFN γ -primed macrophages infected with Tg. These include (1) Live time-lapse imaging of cells infected with Tg which revealed that the macrophages shrink, form membrane bound bodies, (2) full caspase-3/7 activation (new **Figure 2D-E**), (3) nuclear condensation and fragmentation (**Figure EV3H**). These morphological hallmarks, the processing of caspase-substrates, early exposure of annexin-V and late uptake of PI and inhibition with pan-caspase inhibitors together unequivocally indicate apoptotic cell death during Tg infection.*

2. In addition to ASC/Casp-1, ASC/Caspase-8 interaction and cross-talk has also been observed before (Man et al. J. Immunol. 2013, others). Is caspase-8 recruited to ASC specks in caspase-1 deficient cells?

*The reviewer is correct, and we have cited the Man et al study which used STm infection. We have imaged ASC specs in both wild-type and Δ CASP1 cells and find that active caspase-8 (stained with FITC-IETD-fmk) is recruited to ASC specs in both cell types (see new data in **Figure 3C**). About 95% ASC specs are positive for active caspase-8. Therefore, ASC/caspase-8-specks trigger apoptosis during Tg-infection, in a caspase-1-independent manner.*

Does specific inhibition of caspase-8 rescue cells from death (in the presence or absence of caspase-1/4)?

*Yes, in addition to an essential role for caspase-8 in wildtype THP-1 cells (previous Figures 2D-E; now **Figure 2F-G**), its silencing in IFN γ -primed Δ CASP1 or Δ CASP4 cells blocks apoptosis in response (**new Figure EV4A**). Altogether, caspase-8 is essential for cell death induced by Tg and caspase-1/4 are not required.*

3. Pro- and cleaved caspase-1 blots should be shown for unprimed, IFN γ primed, and GBP1-deficient (unprimed and primed) during infection. Just because cells die in the absence of caspase-1, the role for canonical pyroptosis in caspase-1 sufficient cells should be determined in the context of GBP1 presence and absence. The authors should also explain why pro-caspase-1 is decreasing over time during infection (could be as simple as caspase-1 cleavage through canonical AIM2 inflammasome) in Figure 2H and Fig. S3H.

*The lack of cleaved caspase-1 at 2 h post-Tg-infection was shown in the old Figure S3B even though we could detect active caspase-1 with LPS + nigericin as positive control. The 2 h time-post Tg-infection was chosen because pro-caspase-1 is still present at this time point. However, we have now performed similar experiments at 6 hours post-Tg infection without and with IFN γ -priming of WT and THP-1 Δ GBP1 cells (see **new Figure EV3B**). Neither pro-caspase-1 nor cleaved caspase-1 can be detected, supporting our hypothesis that an uncharacterised Tg effector protein might mediate this rather than the processing of caspase-1 by the AIM2 inflammasome.*

Annexin V is also known to label cells undergoing pyroptosis, suggesting it is not a specific marker for apoptosis (Vasconcelos et al. Cell Death and Differentiation 2018).

*We are aware that AnnV can label pyroptotic cells. Importantly, however, our conclusion that cells undergo apoptosis rather than pyroptosis was based on (1) the lack of GSDMD involvement (**Figure 2A**); (2) cleavage and activation of apoptotic caspases and their substrate PARP by immunoblots (**Figure 2C**); (3) live time-lapse imaging showing the activation of apoptotic caspases-3/7 using a fluorescent caspase substrate (**new Figure 2D**).*

4. The cell-type specificity in the recruitment of GBP1 to PCV is interesting. Are there any differences in the replication of *T. gondii* or the structure of PCV between different cell types?

We agree that the effects of GBPs on Tg replication and the ultrastructure of the PV are of great interest. However, comparative analyses of this with two pathogens is beyond scope of this manuscript which focuses on host-cell death. Restriction of microbial replication (a family-wide study across GBPs) will be addressed in a separate manuscript currently under preparation.

Does the cell-type specific recruitment of GBP1 also leads to differences in the mode of cell death in response to *T. gondii* infection?

*This is an interesting point. To address cell-type specific responses, we used A549 cells in which GBP1 does not recruit to Tg vacuoles (Johnston et al, 2016). In agreement with our findings correlating GBP1-targeting to microbes in macrophages, no cell death was observed in response to Tg-infection without or with IFN γ -priming (**new Figure EV4D**). Therefore, recruitment of GBP1 to Tg vacuoles is required for induction of apoptosis.*

5. The authors propose GBP1-mediated release of DNA from the parasite as the preceding event that leads to AIM2-, ASC- and caspase8- dependent cell death. The authors should provide some insights regarding the mechanisms by which GBP1 recruitment to PCV leads to DNA release for sensing by AIM2.

We appreciate this comment but wish to point out that how GBP-recruitment leads to membrane rupture is not clear despite work by many groups with several pathogens for over a decade, especially using murine systems. Our study is the first to clarify that GBPs do not only control pyroptosis but also apoptosis, identifies human GBP1 prenylation and GTPase activities as essential for recruitment to Tg & STm, and promotes the trafficking of caspase-4 to bacteria. The strengths of our study include the use of natural infection models (than mutant strains or microbial ligands), and importantly, primary MDMs and independent loss-of-function approaches (siRNA and CRISPR/Cas9). Altogether, the substantial data we provide make important advances in the field of host-cell death regulated by GBPs during infection by evolutionarily diverse pathogens. While both

our groups agree that questions remain on the mechanism of membrane disruption, these are beyond the scope of this revision and have raised such unanswered questions in the Discussion (lines 397-409).

Minor Comments:

1. Discuss why NLRP1 inflammasome is not involved in mediating cell death to *Toxoplasma* in your infection model, as it has been previously described.

This is an interesting point and may arise from the presence of multiple Nlrp1 genes in rodents versus a single NLRP1 in humans. Further, NLRP1 has been shown to be involved in the human response to Tg infection in monocytes but not in macrophages. We have discussed this in more detail (see lines 421-427).

2. The specificity of the home-made antibody against GBP1 looks good by immunoblot but is not validated for immunofluorescence (Figure 3A, first panels in magenta?). GBP antibodies are notoriously cross-reactive.

We apologise for not showing antibody validation data. As shown in Appendix Figure S2, no staining can be seen with anti-GBP1 in Δ GBP1 cells treated with IFN γ (in which all other GBPs are expressed) which ruled out cross-reactivity. Treatment with Dox alone restores GBP1 signal.

References:

- Antunes LCM, Wang M, Andersen SK, Ferreira RBR, Kappelhoff R, Han J, Borchers CH & Finlay BB (2012) Repression of *Salmonella enterica* *phoP* expression by small molecules from physiological bile. *J. Bacteriol.* 194: 2286–96
- Broz P, Newton K, Lamkanfi M, Mariathasan S, Dixit VM & Monack DM (2010) Redundant roles for inflammasome receptors NLRP3 and NLRC4 in host defense against *Salmonella*. *J. Exp. Med.* 207: 1745–55
- Catron DM, Lange Y, Borensztajn J, Sylvester MD, Jones BD & Haldar K (2004) *Salmonella enterica* serovar Typhimurium requires nonsterol precursors of the cholesterol biosynthetic pathway for intracellular proliferation. *Infect. Immun.* 72: 1036–42
- Franchi L, Amer A, Body-Malapel M, Kanneganti T-D, Ozören N, Jagirdar R, Inohara N, Vandenabeele P, Bertin J, Coyle A, Grant EP & Núñez G (2006) Cytosolic flagellin requires Ipaf for activation of caspase-1 and interleukin 1 β in salmonella-infected macrophages. *Nat. Immunol.* 7: 576–82
- Goddard P, Sanchez-Garrido J, Slater S, Kalyan M, Ruano-Gallego D, Marchès O, Fernández L, Frankel G & Shenoy A (2019) Enteropathogenic *E. coli* stimulates effector-driven rapid caspase-4 activation in human macrophages. *Cell Rep.*: in press
- Henry T, Couillault C, Rockenfeller P, Boucrot E, Dumont A, Schroeder N, Hermant A, Knodler LA, Lecine P, Steele-Mortimer O, Borg J-P, Gorvel J-P & Meresse S (2006) The *Salmonella* effector protein PipB2 is a linker for kinesin-1. *Proc. Natl. Acad. Sci.* 103: 13497–13502
- Henry T, García-Del Portillo F & Gorvel JP (2005) Identification of *Salmonella* functions critical for bacterial cell division within eukaryotic cells. *Mol. Microbiol.* 56: 252–67
- Johnston AC, Piro A, Clough B, Siew M, Virreira Winter S, Coers J & Frickel E-M (2016) Human GBP1 does not localize to pathogen vacuoles but restricts *Toxoplasma gondii*. *Cell. Microbiol.* 18: 1056–64
- Liu BC, Sarhan J, Panda A, Muendlein HI, Ilyukha V, Coers J, Yamamoto M, Isberg RR & Poltorak A (2018) Constitutive Interferon Maintains GBP Expression Required for Release of Bacterial

Components Upstream of Pyroptosis and Anti-DNA Responses. Cell Rep. 24: 155–168.e5

Miao EA, Alpuche-Aranda CM, Dors M, Clark AE, Bader MW, Miller SI & Aderem A (2006) Cytoplasmic flagellin activates caspase-1 and secretion of interleukin 1beta via Ipaf. Nat. Immunol. 7: 569–75

Rathinam VAK, Vanaja SK, Waggoner L, Sokolovska A, Becker C, Stuart LM, Leong JM & Fitzgerald KA (2012) TRIF licenses caspase-11-dependent NLRP3 inflammasome activation by gram-negative bacteria. Cell 150: 606–19

Reyes Ruiz VM, Ramirez J, Naseer N, Palacio NM, Siddarthan IJ, Yan BM, Boyer MA, Pensinger DA, Sauer J-D & Shin S (2017) Broad detection of bacterial type III secretion system and flagellin proteins by the human NAIP/NLRC4 inflammasome. Proc. Natl. Acad. Sci. U. S. A. 114: 13242–13247

Sanchez-Garrido J, Sancho-Shimizu V & Shenoy AR (2018) Regulated proteolysis of p62/SQSTM1 enables differential control of autophagy and nutrient sensing. Sci. Signal. 11: eaat6903

Santos JC, Dick MS, Lagrange B, Degrandi D, Pfeffer K, Yamamoto M, Meunier E, Pelczar P, Henry T & Broz P (2018) LPS targets host guanylate-binding proteins to the bacterial outer membrane for non-canonical inflammasome activation. EMBO J. 37:

Shenoy AR, Wellington DA, Kumar P, Kassa H, Booth CJ, Cresswell P & MacMicking JD (2012) GBP5 Promotes NLRP3 Inflammasome Assembly and Immunity in Mammals. Science (80-.). 336: 481–485

Starr T, Bauler TJ, Malik-Kale P & Steele-Mortimer O (2018) The phorbol 12-myristate-13-acetate differentiation protocol is critical to the interaction of THP-1 macrophages with Salmonella Typhimurium. PLoS One 13: e0193601

Vanaja SK, Russo AJ, Behl B, Banerjee I, Yankova M, Deshmukh SD & Rathinam VAK (2016) Bacterial outer membrane vesicles mediate cytosolic localization of LPS and caspase-11 activation. Cell 165: 1106

Yu X-J, Ruiz-Albert J, Unsworth KE, Garvis S, Liu M & Holden DW (2002) SpiC is required for secretion of Salmonella Pathogenicity Island 2 type III secretion system proteins. Cell. Microbiol. 4: 531–40

2nd Editorial Decision

7th May 2019

Thank you for submitting your revised manuscript to The EMBO Journal. I am sorry for the slight delay in getting back to you with a decision, but have now received all the needed input on the revision.

I had asked both referees #1 and 2 to look at the revised version, but only referee #1 was able to take a look at the revision. The comments by referee #1 are provided in the attached PDF file. As you can see from the comments, the referee appreciates the added revisions, but is more hesitant about the Salmonella infection data. I see the point raised by the referee, but also find that the manuscript adds important insight and that you have done a good job to responding to the referees' concerns.

I have also involved an external advice to look at the revised manuscript and the issues raised by the referee who was also in agreement with this view and supported publication here.

Regarding the remaining concerns raised by referee #1: Please make sure that you carefully discuss the caveats that the referee raises. When you upload the revised version please also provide a point-by-point response.

When you submit the revised version will you also take care of the following issues:

REFEREE REPORT:

Referee #1:

Report sent as word file to editor.

Please see next page.

In this revised MS the authors have addressed all, but not answered some, of my remarks on the *Salmonella* data. Unfortunately my major criticism remains about the physiological relevance of the data with respect to *Salmonella* infection. Perhaps I was not sufficiently clear in my initial report so to aid the authors such that they understand my concerns I have highlighted to them the problem on the figure below taken from their revised paper and added a red oval to clarify my comments. The authors have focussed on my comments on the overall levels of cell death (as a point of interest many researchers do get rapid NAIP-dependent cell death in a high % of human macrophages in response to infection with *Salmonella*, for example Kortmann et al; <https://doi.org/10.4049/jimmunol.1403100> as well as seeing other forms of cell death), but my major concern was with the difference between the IFN γ stimulated and unstimulated cells as highlighted below. Data may be statistically significant but that does not always mean they are biologically relevant (for an interesting discussion on this point I refer the authors to a recent commentary in Nature <https://www.nature.com/articles/d41586-019-00857-9>).

□

Biological relevance would require mouse data and, whilst I appreciate the problems with this given the proposed human-mouse differences, this leaves the bacterial section of this paper in difficulty. The problems with this MS are three fold. Firstly the authors assert that NAIP-NLRC4 driven cell death is relatively unimportant in human macrophage so their assertions that caspase 4 driven cell death is highly important are not supported by the literature (although NAIP-driven cell death may be limited in THP-1 cells, this is not the case in primary human cells Kortmann et al; <https://doi.org/10.4049/jimmunol.1403100>). Secondly the small amount of cell death driven by GBP1 leaves questions about the biological significance of these data. Thirdly caspase 11 driven cell death is well known to utilise GBPs (doi: [10.1073/pnas.1321700111](https://doi.org/10.1073/pnas.1321700111) doi:10.1038/nature13157 doi: [10.1073/pnas.1321700111](https://doi.org/10.1073/pnas.1321700111)) including during bacterial infection. The authors are to be commended on using human cells and wild type bacteria in a physiological system, but currently there are many questions remaining due to the limited effect of GBP1 in this MS for the *Salmonella* data.

The authors performed the requested editorial changes.

Response to Reviewer 1.

We thank the reviewer for their comments. We are glad that they are largely happy with our new experiments and find most of our work interesting. However, we are surprised to see them question the "physiological relevance", especially given previous work on IFN-mediated elevation of pyroptosis during infection e.g. Santos et al EMBO J. 2018 Mar 15;37(6). pii: e98089. and Liu et al, Cell Rep 2018 Jul 3;24(1):155-168.e5. We also note that the reviewer mainly addresses STm-related work and has missed the importance of our overall findings on GBP1 which provide new mechanistic insight on the responses of human macrophages to infection by two medically relevant pathogens. Reviewer's text appears in *blue italics* in our point-by-point response below.

1. *"Data may be statistically significant but that does not always mean they are biologically relevant (for an interesting discussion on this point I refer the authors to a recent commentary in Nature <https://www.nature.com/articles/d41586---019---00857---9>)." (we assume this is the article, which we were aware of, as the link from reviewer is broken <https://www.nature.com/articles/d41586-019-00857-9>).*

We agree and are very surprised the reviewer cites this article which talks about *NOT* relying on statistics to infer biological relevance! The article states: *"Whatever the statistics show, it is fine to suggest reasons for your results, but discuss a range of potential explanations, not just favoured ones. Inferences should be scientific, and that goes far beyond the merely statistical."* Indeed, in our manuscript we have not relied solely on statistics. The biological relevance of IFN γ -mediated responses in STm and Tg infections is well appreciated from previous work in humans and mice as we outlined in the Introduction.

2. The reviewer is incorrect in saying that *"Firstly the authors assert that NAIP-NLRC4 driven cell death is relatively unimportant in human macrophage..."*, because we have made no such claim. Indeed, our schematic figure shows NAIP-NLRC4 inflammasomes. In the first paragraph of our discussion we state that *"Our results highlight the contribution of IFN γ priming, the host-species and microbial pathogen to macrophage cell death."*; *"We also found that during STm infection, IFN γ enhanced macrophage pyroptosis in a GBP1-dependent manner."* This is consistent with our experimental results and does not downplay any previous work. We have now cited both Kortmann et al and Reyes-Ruiz et al to satisfy the reviewer. Further, because we did *NOT* prime MDMs with LPS (as is done in the Kortmann study cited by the reviewer), we could see lesser absolute levels of death. STm is a Gram-negative pathogen and introduces LPS itself during infection. LPS-priming is physiologically irrelevant for our side-by-side assays with Tg. Our experimental conditions are therefore scientifically justified. We would also like to point out that neither Kortman et al nor Reyes Ruiz et al (10.1073/pnas.1710433114) claim that NAIP is involved in detecting natural infection by wild type STm. They do, however, confirm that the cytosolic delivery of STm needle, rod or flagellin proteins activates the human NAIP-NLRC4 inflammasome. Furthermore, neither study used IFN γ -stimulated macrophages, which is the central focus of our study.

3. *"although NAIP-driven cell death may be limited in THP-1 cells, this is not the case in primary human cells Kortmann et al;..."* As described above, NAIP-driven cell death by STm has not been unequivocally shown in human MDMs by Kortmann et al and Reyes Ruiz et al, and neither study used IFN γ -stimulated cells. Therefore, those studies cannot be compared to our findings on GBP1.

4. *"Secondly the small amount of cell death driven by GBP1 leaves questions about the biological significance of these data."*

We respectfully disagree. Also see response to point 2 above. Importantly, our findings with THP-1 cells are consistent with those with primary MDMs. The reviewer also contradicts themselves by questioning the biological importance and then (i) pointing to above commentary and (ii) admitting that *"Biological relevance would require mouse data ... I appreciate the problems with this given the proposed human-mouse differences..."*. Differences in humans and mice are not 'proposed' and are a fact – humans have 7 GBPs 1 NAIP, 2 LPS-binding caspases (caspase-4 & -5) whereas mice have 12 GBPs, 4 functional Naips and one caspase-11.

5. *"Thirdly caspase 11 driven cell death is well known to utilise GBPs"*

We do not dispute this and have cited previous work on mouse GBPs. Note that type I IFNs, which upregulate mGBPs and caspase-11 *in vitro* through autocrine LPS-signalling in murine cells are not known to do so in human cells. Human cells need type II IFN (IFN γ) to upregulate GBP expression *in vitro* and human caspase-4 is constitutively expressed.

6. *“The authors are to be commended on using human cells and wild type bacteria in a physiological system, but currently there are many questions remaining due to the limited effect of GBP1 in this MS for the Salmonella data.”*

We thank the reviewer for appreciating our efforts. We want to reiterate that we find GBP1 is responsible for IFN γ -driven enhancement of host cell death in STm-infected human primary and THP-1 macrophages. We also agree that several questions remain unanswered, especially with respect to naïve macrophages, which is natural for research like this and we hope we and others will continue to address them in future work. For instance, our Discussion also says “...*human caspase-4 can be activated in naïve macrophages infected with Francisella (Lagrange et al., 2018) or enteropathogenic E. coli (Goddard et al., 2019), and the absence of a major role for caspase-4 in naïve macrophages infected with STm needs to be investigated further in the future.*”

Lastly, we request the reviewer to take our manuscript in its entirety. We used two pathogens and uncovered that GBPs not only participate in non-canonical inflammasome signalling, but also in atypical apoptosis which is a major advance on previous work.

3rd Editorial Decision

13th May 2019

Thank you for submitting your revised version. I have now had a chance to take a look at everything and all looks good. I am therefore very pleased to accept the manuscript for publication here.

Corresponding Author Name: Eva-Maria Fricke & Avinash Shenoy

Manuscript Number: EMBOJ-2018-100926